# Mapping replication dynamics in *Trypanosoma brucei* reveals a link with telomere transcription and antigenic variation

Rebecca Devlin[†], Catarina A Marques[†], Daniel Paape, Marko Prorocic, Andrea C Zurita-Leal, Samantha J Campbell, Craig Lapsley, Nicholas Dickens, Richard McCulloch*

The Wellcome Trust Centre for Molecular Parasitology, Institute of Infection, Immunity and Inflammation, University of Glasgow, Glasgow, United Kingdom

**Abstract** Survival of *Trypanosoma brucei* depends upon switches in its protective Variant Surface Glycoprotein (VSG) coat by antigenic variation. VSG switching occurs by frequent homologous recombination, which is thought to require locus-specific initiation. Here, we show that a RecQ helicase, RECQ2, acts to repair DNA breaks, including in the telomeric site of VSG expression. Despite this, RECQ2 loss does not impair antigenic variation, but causes increased VSG switching by recombination, arguing against models for VSG switch initiation through direct generation of a DNA double strand break (DSB). Indeed, we show DSBs inefficiently direct recombination in the VSG expression site. By mapping genome replication dynamics, we reveal that the transcribed VSG expression site is the only telomeric site that is early replicating – a differential timing only seen in mammal-infective parasites. Specific association between VSG transcription and replication timing reveals a model for antigenic variation based on replication-derived DNA fragility.

*For correspondence: Richard. McCulloch@glasgow.ac.uk

[†]These authors contributed equally to this work

Competing interests: The authors declare that no competing interests exist.

## Introduction

The growth and propagation of pathogens in vertebrates requires strategies to survive the host immune responses, in particular adaptive immunity. One such survival strategy, found widely in biology, is antigenic variation, which involves periodic switches in exposed pathogen antigens, thereby allowing a fraction of the infecting population to escape immune clearance. A number of strategies for antigenic variation have been described, though normally only one is employed in any given pathogen. In this regard, antigenic variation in the African trypanosome, *Trypanosoma brucei*, is unusual, since here two apparently distinct approaches are adopted: recombination and transcription. Antigenic variation in *T. brucei* involves switches in the identity of the Variant Surface Glycoprotein (VSG) expressed on the cell surface, where the protein forms a dense 'coat' that is believed to shield invariant antigens from immune recognition (*Higgins et al., 2013*). At any given time an individual *T. brucei* cell in the mammal expresses only one *VSG* gene, due to transcriptional control mechanisms that ensure only one of ~15 *VSG* transcription sites, termed bloodstream expression sites (BES), is active. Such monoallelic expression is found in other antigenic variation systems, such as that involving the ~60 *var* genes in *Plasmodium falciparum* (*Guizetti and Scherf, 2013*), as is the ability to switch the gene that is actively transcribed, eliciting antigenic variation. The nature of the monoallelic control and transcriptional switch mechanisms in *T. brucei*, and whether they share features with other pathogens, are still being unraveled (*Horn, 2014*). One complexity in understanding

**eLife digest** The African trypanosome, *Trypanosoma brucei,* is a parasite that is transmitted between mammals by the tsetse fly, and causes a disease known as sleeping sickness in humans. Like many other parasites, trypanosomes have evolved ways to avoid being killed by their hosts. One such survival strategy involves the parasites constantly changing the molecules that coat their surface, which are the main targets recognized by their hosts' immune systems. Switching one coat protein for another similar protein, a process called antigenic variation, allows a parasite to evade an attack and establish a persistent infection. Antigenic variation also makes it almost impossible to develop a vaccine that will offer lasting protection against the parasite.

Previous research suggested that a trypanosome might deliberately break its own DNA and then exploit a repair process to switch its current coat protein-encoding gene for another one located elsewhere within its genetic material. Devlin, Marques et al. now reveal that it is unlikely that trypanosomes use a specific enzyme to break DNA deliberately during coat switching. Instead, experiments using whole-genome sequencing suggest that coat-gene-switching might arise from the strategies trypanosomes use to copy their genetic material during cell division.

These findings bring researchers closer to understanding how trypanosomes start antigenic variation in order to evade their hosts' immune responses. In addition, the findings suggest a new model that could help researchers answer an important question: how does the timing of genome copying vary from cell to cell? Nevertheless, the hypothesis proposed by Devlin, Marques et al. will now require rigorous testing. Future studies could also ask if other parasites use similar strategies to survive being attacked by their host's immune systems.

VSG transcriptional switching is the elaborate structure of the BES (*Hertz-Fowler et al., 2008*), where the *VSG* is co-transcribed with many other genes, termed expression site-associated genes (ESAGs), from an RNA Polymerase I promoter. Despite some variation in *ESAG* composition between BES, two features appear invariant in all these sites: the *VSG* is always proximal to the telomere and is separated from the upstream *ESAG*s by an array of 70 bp repeats (which appear to be only found adjacent to *VSG*s in the *T. brucei* genome)(*Marcello and Barry, 2007*). Transcriptional switching occurs between the *VSG*s that occupy the BES, and is therefore limited by BES number. However, a second route of VSG switching relies upon recombination and can over-write the BES-resident *VSG*s, generating new VSG coats from a genomic archive of ~2000 *VSG*s (*Cross et al., 2014*; *Marcello and Barry, 2007*). In numerical terms, therefore, recombination is the major route for VSG switching. Indeed, recombination is a very widespread strategy for antigenic variation in eukaryotic and bacterial pathogens (*Palmer and Brayton, 2007*), most likely because it drives antigen diversity, which prolongs infection and facilitates transmission (*Hall et al., 2013*; *Mugnier et al., 2015*).

The *VSG* archive is distributed across the three chromosome classes that comprise the *T. brucei* nuclear genome. A small part of the archive is the BES (*Hertz-Fowler et al., 2008*), which are found in the 11 diploid megabase chromosomes as well as in the ~5 aneuploid intermediate chromosomes. A larger part of the archive is found at the telomeres of ~100 minichromosomes (*Wickstead et al., 2004*), where *ESAG*s and BES promoters have not been found, suggesting this part of the archive is simply a store of silent *VSG*s for recombination. The largest silent store is composed of arrays of *VSG*s in the subtelomeres of the megabase chromosomes, where the majority of the *VSG*s are pseudogenes or partial genes (*Berriman et al., 2005*). The strategies for *VSG* recombination in antigenic variation reflect the archive location and gene composition (*McCulloch et al., 2015*). A minor route for switching is termed reciprocal *VSG* recombination, where telomeres are exchanged between two chromosomes, moving the *VSG* out of the active BES and moving a previously silent *VSG* into the active BES (*Rudenko et al., 1996*). More common is *VSG* gene conversion, which can involve both intact and impaired *VSG*s, and involves deletion of the *VSG* in the BES and replacement by *VSG* sequence copied from the silent archive. Early in infections gene conversion of intact *VSG*s predominates (*Marcello and Barry, 2007*; *Morrison et al., 2005*) and, since the *VSG*s share little sequence homology, the reaction relies on flanking homology. ~90% of *VSG*s are flanked by 70 bp repeats

(*Marcello and Barry, 2007*), which provide upstream homology to guide recombination of virtually all genes in the archive. In addition, gene conversion of *VSG*s between BES can use extensive upstream homology: gene conversion can extend to downstream homology within and around the VSG open reading frame (ORF) or, if the silent *VSG* is telomeric, to the chromosome end. Impaired *VSG*s are seen as recombination substrates later in infections and here gene conversion differs from intact *VSG*s, since the reaction involves the production of a functional gene using homology within the ORF; indeed, multiple *VSG* donors are frequently recombined to generate novel 'mosaic' *VSG*s in a reaction termed segmental gene conversion (*Hall et al., 2013*; *Mugnier et al., 2015*).

All available evidence suggests switching of intact *VSG*s by recombination is catalyzed by homologous recombination (HR), a universally conserved reaction that directs repair of DNA damage and maintains replication fork progression genome-wide. Mutation of the central catalytic enzyme of HR, RAD51, impairs (but does not abolish) VSG recombination, including gene conversion (*McCulloch and Barry, 1999*). Consistent with that phenotype, mutation of *T. brucei* BRCA2 (*Hartley and McCulloch, 2008*) and at least one of four RAD51 paralogues (*Dobson et al., 2011*) - factors that aid RAD51 function - has the same outcome. More recently, mutation of TOP3α or RMI1, which interact and may be components of the *T. brucei* RTR (RecQ/Sgs1-Top3/TOPO3α-Rmi1/BLAP75/18) complex (*Mankouri and Hickson, 2007*), was shown to result in increased VSG switching, an effect that is RAD51-dependent (*Kim and Cross, 2010*; *2011*). The conclusion that antigenic variation can be executed by a non-specific, general repair pathway is not limited to *T. brucei*, as similar gene knockout studies in *Neisseria gonorrhoeae* implicate HR in pilin antigenic variation (*Cahoon and Seifert, 2011*). However, VSG switching can occur at rates substantially higher than might be predicted for background mutation (*Turner, 1997*) and may be focused to target the active BES, features that may suggest some mechanistic specialization or locus-specificity. As a result, recent work has explored how VSG switching might be initiated in *T. brucei*, leading to an association between elevated rates of switching and DNA double strand breaks (DSBs). The evidence for this association is two-fold. First, controlled induction of the endonuclease I-SceI to specifically generate a DSB adjacent to the 70 bp in the active BES leads to a ~250 fold increase in VSG switching by recombination (*Boothroyd et al., 2009*), an effect not seen when a DSB is induced in other locations in the active BES (*Boothroyd et al., 2009*; *Glover et al., 2013*) or when the 70 bp repeats have been deleted from the active BES (*Boothroyd et al., 2009*). Second, ligation-mediated PCR is able detect DNA breaks in the BES, with the lesions initially reported to be limited to the vicinity of the 70 bp repeats in the active BES (*Boothroyd et al., 2009*), though later also reported in the silent BES (*Glover et al., 2013*; *Jehi et al., 2014*) and found to be more widely distributed in the transcription units (*Glover et al., 2013*).

Despite the emerging association between DNA DSBs and VSG switching, questions remain about the detailed mechanism(s) of VSG switch initiation. For instance, are DSBs generated directly in the active BES, such as through the action of an endonuclease, as occurs during *Saccharomyces cerevisiae* mating type switching (*Lee and Haber, 2015*)? Alternatively, might other processes lead more indirectly to break formation and elicit switching, such as the transcription, replication and DNA nicking events that initiate locus-directed recombination reactions during, respectively, immunoglobulin gene switching in mammals (*Roth, 2014*), mating type switching in *Schizosaccharomyces pombe* (*Klar et al., 2014*) and pilin antigenic variation in *N. gonorrhoeae* (*Obergfell and Seifert, 2015*)? In this study, we have examined VSG switch initiation in two ways. First, we describe the impact on DNA repair and VSG switching caused by mutation of one of two *T. brucei* RecQ-like helicases, which we have named TbRECQ2. We show that loss of TbRECQ2 impairs DSB repair, consistent with the observation that the protein localizes to such lesions. Conversely, TbRECQ2 mutants display elevated rates of VSG switching, indicating it is unlikely that the direct formation of DSBs is the initiating event in VSG switching. Second, we provide evidence for strong association between replication timing and BES transcription, indicating that VSG switch initiation may be mechanistically linked to DNA replication.

# Results

## T. brucei RECQ2 is non-essential in bloodstream form cells and acts in DNA damage repair

Helicases are molecular motors that use energy released from nucleoside triphosphate hydrolysis to unwind RNA, DNA or RNA:DNA hybrids. RecQ-like helicases are widespread Superfamily 2 DNA helicases (*Fairman-Williams et al., 2010*), identifiable by homology with the first RecQ helicase described in *Escherichia coli* (*Umezu et al., 1990*). *S. pombe* and *S. cerevisiae*, single-celled yeast, encode a single RecQ helicase: Rqh1 and Sgs1, respectively. In contrast, multicellular eukaryotes such as humans and *Drosophila menaogaster* possess multiple RecQ helicases, with five discernible in mammals (*Bernstein et al., 2010*; *Hickson, 2003*). RecQ helicases function in diverse aspects of genome maintenance. For example, human RECQ4 interacts with the replication factors MCM10 and MCM2-7, while RECQL4 interacts with CDC45 and GINS (*Im et al., 2009*; *Xu et al., 2009*). RecQ helicases also play roles in non-homologous end-joining, since human WRN interacts with XRCC4-ligase IV (*Kusumoto et al., 2008*), while human RECQ1 binds Ku70/80 and its depletion leads to reduced repair (*Parvathaneni et al., 2013*). Finally, RecQ helicases act in both the initiation and execution of HR (*Haber, 2015*), since the RTR complex promotes DNA DSB resection (*Mimitou and Symington, 2008*; *Zhu et al., 2008*), controls DNA annealing during strand invasion (*Fasching et al., 2015*; *Spell and Jinks-Robertson, 2004*), and 'dissolves' Holliday junction intermediates to limit crossover genetic exchange between chromosomes (*Cejka et al., 2010*; *Hickson and Mankouri, 2011*).

BLAST searches with multiple eukaryotic RecQ helicase sequences consistently revealed two well-aligned proteins encoded in the *T. brucei* genome, which we arbitrarily named TbRECQ1 (TriTrypDB gene accession number Tb427.06.3580) and TbRECQ2 (Tb427.08.6690). RNAi analysis suggests that TbRECQ1, which appears more distantly related to eukaryotic RecQ helicases than TbRECQ2, is essential (Devlin et al, unpublished). Protein domain predictions (*Figure 1A*) suggest that TbRECQ2 contains a conserved DEAD/DEAH box helicase domain, indicating potential ATP and nucleic acid binding activity, and a more C-terminal helicase domain that is found in helicases of multiple families (*Linder, 2006*). In addition, an HRDC (helicase and RNaseD C-terminal) domain is predicted close to the C-terminus. In contrast to the two other domains, the HRDC domain appears limited to some RecQ helicases and RNase D homologues (*Morozov et al., 1997*), where it is probably involved in DNA binding (*Bachrati and Hickson, 2003*). However, HRDC domains are not found in all RecQ helicases; for example, three human RecQ helicases, WRN, BLM and RECQ1, each contain an HRDC, but it is absent in human RECQ4 and RECQ5 (*Bernstein et al., 2010*).Thus, the prediction that TbRECQ2 contains an HRDC might suggest a function closer to the former human RecQ helicases. One domain that is limited to RecQ helicases is termed RQC (RecQ C-terminal), which may be involved in protein-protein interactions (*Bernstein et al., 2010*), as well as in binding and unwinding dsDNA at branch points (*Kitano et al., 2010*). Though this domain could not be predicted in TbRECQ2, it is also absent from or highly diverged in some other validated RecQ proteins (*Bachrati and Hickson, 2003*) and the evolutionary distance between *T. brucei* and the most characterised model eukaryotes may confound identification.

To evaluate the role of TbRECQ2 in *T. brucei* bloodstream form (BSF) cells, heterozygous (+/-) and homozygous (-/-) knockout mutants were generated by sequential transformation with constructs that replace nearly all of the *TbRECQ2* ORF with cassettes that express resistance to blasticidin or G418 (*Figure 1—figure supplement 1A*). Integration of the constructs and deletion of both wild type (WT) *TbRECQ2* alleles in the -/- mutants was confirmed by PCR (*Figure 1—figure supplement 1B*), while reverse transcription PCR (RT-PCR) (*Figure 1—figure supplement 1C*) showed no intact *TbRECQ2* mRNA could be detected in the -/- mutants. The successful generation of null mutants shows TbRECQ2 is not essential in BSF *T. brucei*, though growth analysis revealed that the *recq2-/-* mutants had an increased population doubling time compared with WT cells (*Figure 1B*). Indeed, slowed growth was also apparent in the *req2+/-* mutants, suggesting a growth impediment after loss of one allele that becomes more severe in the null mutant. To ask if the growth change results from an impediment in completing a cell cycle stage or traversing between sequential stages, cells were stained with DAPI to visualise nuclear (N) and kinetoplast (K) DNA. Counting the relative numbers of the two *T. brucei* genomes allows a cytological assessment of the cell cycle stage of individual cells in the population (*McKean, 2003*), and we found no change in the proportion of 1N1K

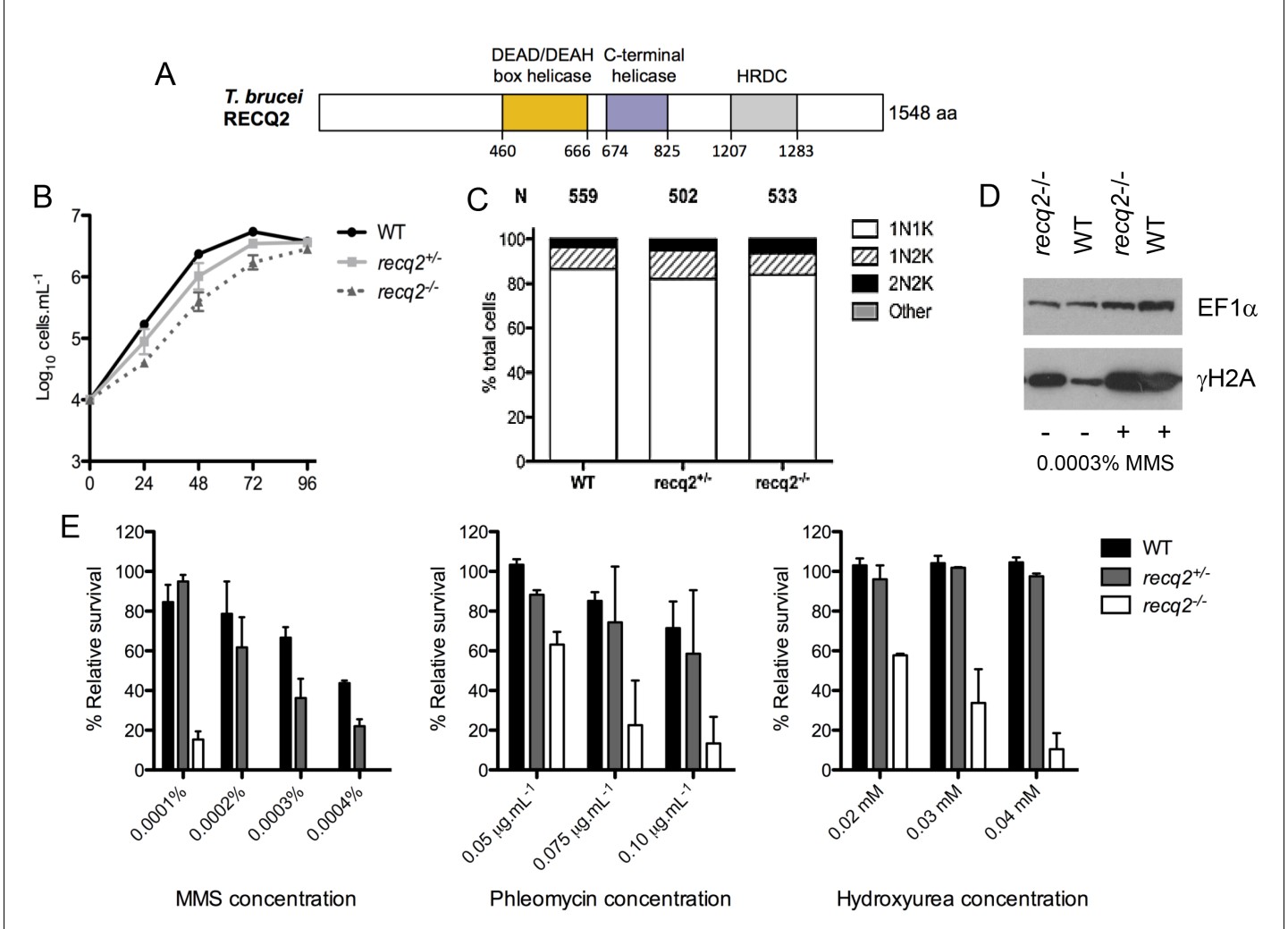

**Figure 1.** *T. brucei* RECQ2 is non-essential and acts in genome repair. (**A**) Representation of predicted protein domains in TbRECQ2. Approximate position (in amino acids, aa, from the N-terminus, N) of predicted functional domains (boxed) is shown underneath the diagram (not to scale). (**B**) The cell density of wild type (WT) cells and *recq2* +/- and -/- mutants cultures was counted every 24 hr up to a maximum of 96 hr, starting from a cell density of 1 x 10$^4$ cells.mL$^1$. The mean cell density from three independent experiments is shown on a Log$_{10}$ Y-axis graph; error bars depict standard error of the mean. (**C**) Cell cycle analysis of WT cells and *recq2* +/- and -/- mutants. DNA content was evaluated after DAPI staining of fixed cells and the number of cells with one nucleus and one kinetoplast (1N1K, white box), one nucleus and two kinetoplasts (1N2K, hatched box), two nuclei and two kinetoplasts (2N2K, black box) and cells that did not fit into any of these categories (other, grey box) were counted. The proportion of each cell type is represented as a percentage of the total cells counted (N).(**D**) Western blotting of whole cell extracts from WT and *recq2-/-* mutants grown in the absence (-) of methyl methane sulphonate (MMS), or for 18 hr in media containing 0.0003% MMS (+). Blots were probed with peptide antiserum recognizing Thr130 phosphorylated *T. brucei* histone H2A (γ-H2A) and, as loading control, polyclonal antiserum recognizing *T. brucei* EF1α. (**E**) Clonal survival ofwild type (wt) cells and *recq2* heterozygous (+/-) or homozygous (*recq2-/-*) mutants is shown in the presence of varying concentrations of MMS, phleomycin or hydroxyurea (HU). Mean survival (%) is plotted of the treated cells relative to untreated from three independent experiments, with vertical lines representing standard error of the mean.

The following figure supplement is available for figure 1:

**Figure supplement 1.** Generation of *recq2* null mutants in bloodstream form *T. brucei*.

(G1-S phase), 1N2K (G2-M phase) or 2N2K (post M phase) cells in mutants relative to WT (*Figure 1C*). Thus, the growth impairment of the RECQ2 mutants is not due to detectable stalling at a discernible cell cycle stage or transition. Western blotting to detect the levels of Thr130 phosphorylated histone H2A (γ-H2A) revealed an increased signal in the *recq2-/-* mutants relative to WT (*Figure 1D*), indicating this modification accumulates in the absence of the helicase to an extent

comparable to that seen in WT or -/- mutants cells grown for 18 hr in 0.0003% MMS (see below). This form of phosphorylation of histone H2A is a modification seen after various genotoxic treatments in *T. brucei* (*Glover and Horn, 2012*), suggesting it is the kinetoplastid variant of a conserved eukaryotic chromatin alteration that acts a prelude to repair. Thus, accumulation of the histone variant in the *recq2-/-* mutants indicates an increased level of nuclear DNA damage, which appears not to impair cell cycle progression but may impede cell growth or survival.

To ask if TbRECQ2 contributes to genome repair in *T. brucei* survival of the mutants was compared with WT cells following exposure to three DNA damaging compounds: hydroxyurea (HU), methyl methanesulfonate (MMS) and phleomycin. HU depletes the cellular dNTP pool (*Bianchi et al., 1986*), resulting in stalled replication forks that can subsequently collapse and generate DNA breaks. MMS methylates purines (*Brookes and Lawley, 1961*), which causes DNA breaks, at least in part through DNA repair activities targeting the alkylation (*Lundin et al., 2005*; *Wyatt and Pittman, 2006*). MMS damage also perturbs replication, due to alkylated nucleotides blocking replication fork progression (*Groth et al., 2010*). Phleomycin blocks the activity of DNA polymerase, inhibiting DNA synthesis and resulting in the formation of, primarily, DNA DSBs (*Falaschi and Kornberg, 1964*; *Reiter et al., 1972*). *Figure 1E* shows clonal survival assays, which revealed that the *recq2-/-* cells displayed a much increased sensitivity to MMS compared with WT. Indeed, at MMS concentrations at and above 0.0002%, *recq2+/-* survival was lower than WT. These data are consistent with the increased MMSsensitivity previously reported in *S. cerevisiae* SGS1 mutants (*Mullen and Brill, 2000*), as well as human and chicken DT40 *blm-/-* mutants (*Imamura et al., 2001*), and indicate TbRECQ2 is involved in the *T. brucei* response to MMS-induced damage. Clonal survival showed that TbRECQ2 also contributes to the *T. brucei* response to phleomycin and HU damage, since in both cases the *recq2-/-* mutants were more sensitive than WT cells (*Figure 1E*). However, in neither case was there clear evidence that the *recq2+/-* cells were more sensitive than WT, perhaps indicating a more pronounced role for the putative helicase in tackling MMS damage.

## *T. brucei* RECQ2 localises in nuclear foci after the induction of DNA breaks

To examine the subcellular localisation of TbRECQ2, the protein was N-terminally tagged with 12 copies of the myc epitope (12myc) using a modified version (gift, A.Trenaman) of the pEnT6B construct (*Kelly et al., 2007*), which allowed the variant protein to be expressed from the endogenous *TbRECQ2* locus. A western blot of a cell lysate from a transformant clone showed expression of a myc-tagged protein of the expected size (182 kDa; *Figure 2A*). To test the functionality of the 12myc-TbRECQ2 variant, the untagged *TbRECQ2* allele was deleted by replacement with a G418-resistance cassette (*Figure 1—figure supplement 1A*). MMS sensitivity of the resulting *recq2* 12myc/- cells was then assessed by clonal survival (*Figure 2B*). As survival of the cells expressing only the 12myc tagged variant of TbRECQ2 was comparable with WT cells and *recq2+/-* mutants in the presence of MMS, and notably better than *recq2-/-* mutants, addition of the epitope does not impair TbRECQ2 function in repair.

Localisation of 12myc-TbRECQ2 was examined by direct immunofluorescence with anti-myc antiserum conjugated with Alexa-Fluor 488 (*Figure 2C*). In most cells no signal could be detected, though in the very small proportion (0.2%) that did show a signal (*Figure 2D*), this was seen as a discrete puncta in the nucleus (data not shown). Such a pattern is reminiscent of localisation described for *T. brucei* RAD51 (*Dobson et al., 2011*; *Glover et al., 2008*; *Hartley and McCulloch, 2008*; *Proudfoot and McCulloch, 2005*; *Trenaman et al., 2013*), which is normally not seen in the cell, but localises in what have been described as foci in a small number of cells in the absence of induced damage. RAD51 foci are thought to be repair-related structures, as their numbers increase after damage, both in *T. brucei* and in many other cells (*Bergink et al., 2013*; *Haaf et al., 1995*; *Tarsounas et al., 2004*). As a result, we examined co-localisation of 12myc-TbRECQ2 and TbRAD51, detecting the latter by indirect immunofluorescence with polyclonal anti-RAD51 antiserum. To ask if the 12myc-TbRECQ2 signals might represent repair–related foci and, indeed, might be structurally associated with TbRAD51 foci, the cells were treated for 18 hr with phleomycin at 1 μg.mL$^{-1}$, which has been shown to generate a majority of cells with TbRAD51 foci (*Dobson et al., 2011*; *Hartley and McCulloch, 2008*; *Trenaman et al., 2013*). TbRAD51 foci were observed in ~2% of untreated cells (*Figure 2D*), which is similar to previous studies and comparable with 12myc-

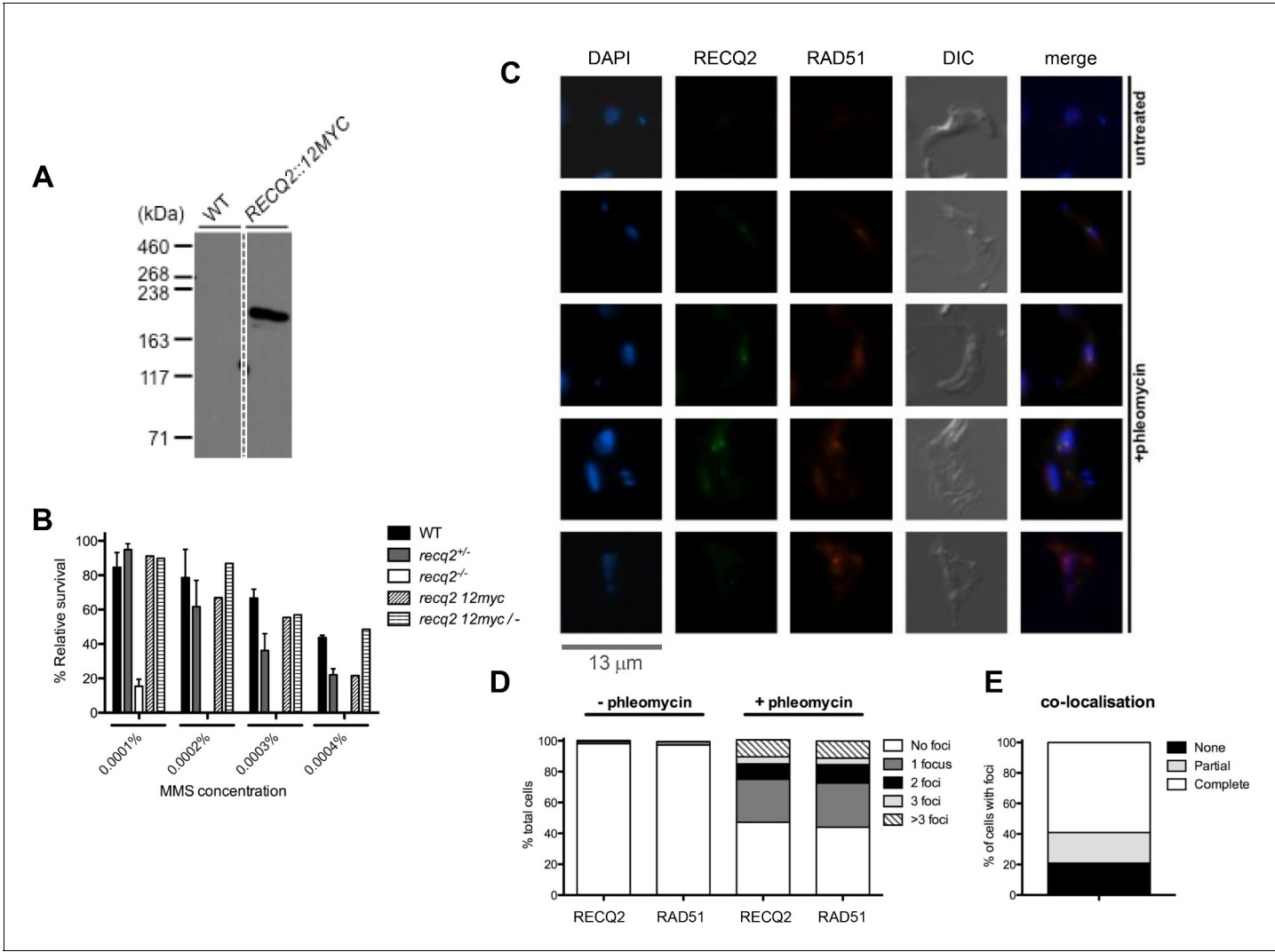

**Figure 2.** TbRECQ2 is a nuclear factor that relocalises to foci in the presence of DNA damage, when it colocalises with RAD51. (A) Western blot analysis of WT cells relative to cells expressing 12myc-tagged RECQ2 from the endogenous locus; size markers are shown (kDa). (B) Clonal survival of cells expressing myc-tagged RECQ2 (12myc and 12myc/-) is shown relative to WT, *recq2+/-* and *recq2-/-* cells in the presence of varying concentrations of MMS. Mean survival (%) is plotted of the treated cells relative to untreated from three independent experiments, with vertical lines representing standard error of the mean. (C) Representative examples of 12myc-RECQ2 and RAD51 cellular localisation in fixed cells, including after 18 hr growth in the presence or absence of phleomycin (1 μg.mL-1); bar: 13 μm. The tagged protein was detected by direct immunofluorescence using an anti-myc antiserum coupled with the Alexa Fluor 488 flurophore (myc, green), while RAD51 was localised by indirect immunofluorescence using a rabbit anti-RAD51 antisera and an Alexa Fluor 594 goat anti-rabbit IgG antiserum; DNA was visualized with DAPI, and differential interference contrast (DIC) was used to visualise whole cells. (D) Percentage of cells containing 12myc-TbRECQ2 and RAD51 foci, as well as the number of detectable foci either in the absence (- phleomycin) or presence (+ phleomycin) of phleomycin is shown. (E) Cells containing 12myc-RECQ2 and RAD51 foci following phleomycin treatment were categorised according to the degree of foci co-localisation, represented as percentage of cells that contained foci.

TbRECQ2. After growth in phleomycin, 47% of cells contained one or more 12myc-TbRECQ2 foci and 44% of cells contained one or more TbRAD51 foci (*Figure 2D*). Indeed, not only was the relative proportion of cells with 12myc-TbRECQ2 and TbRAD51 foci comparable in these conditions, but the pattern of foci accumulation was highly related: most cells contained a single focus of either protein, though some contained 2 or 3 discrete foci, while others had larger numbers (difficult to count accurately). Moreover, there was substantial overlap in the two signals, examples of which are shown in *Figure 2C*. In the majority of cells in which 12myc-TbRECQ2 and TbRAD51 foci were seen (irrespective of the number of foci), the signals co-localised fully (~60% of cells with both foci, *Figure 2E*). In ~20% of cells with foci, the signal overlap was partial because the numbers of 12myc-TbRECQ2 and

TbRAD51 foci were not equivalent in a single cell. Finally, in ~20% of cells there was no overlap, because 12myc-TbRECQ2 displayed foci but TbRAD51 did not, or vice versa. In summary, there appears to be pronounced similarity in behaviour and overlap in signal between 12myc-TbRECQ2 and TbRAD51 before and after phleomycin- induced damage. Whether the non-overlapping signals merely reflect incomplete resolution of one or other signal, or tell us that the proteins can act in subtly different manners (perhaps temporally or spatially), is unclear.

## Loss of TbRECQ2 impairs survival of *T. brucei* cells after DNA double strand break induction

In order to test directly if TbRECQ2 acts in DNA DSB repair, we utilised two cell lines (*Figure 3A*) in which a single DNA DSB can be controllably induced in the genome (gift, David Horn). Both the HR1 (or INT) (*Glover and Horn, 2014*; *Glover et al., 2008*) and HRES (or TEL, VSG$^{up}$)(*Glover et al., 2013*; *Glover and Horn, 2014*) cells have been modified such that expression of the I-SceI meganuclease is dependent upon addition of tetracycline (Tet) to alleviate transcriptional repression by the Tet repressor. HR1 and HRES cells differ in the location of the I-SceI recognition site (*Figure 3A*). In HR1, the I-SceI site is located on chromosome 11, between genes Tb927.11.4530 and Tb927.11.4540 (tritrydb.org), and >1 Mbp from the nearest telomere. Here, the I-SceI site is embedded within an *RFP* (red fluorescent protein):*PUR* (puromycin N-acetyl transferase) fusion gene. The *RFP:PUR* gene is flanked by tubulin sequences for mRNA transplicing and polyadenylation. Thus, HR-directed repair after I-SceI-induced DSB formation could occur by recombination between chromosome 11a (containing the I-SceI site) and its homologue (11b), but could also occur ectopically with chromosome 1 (where the tubulin locus is found) using the short tubulin sequences on the *RFP: PUR* cassette (*Glover et al., 2008*). In HRES, the I-SceI recognition site is located upstream of *VSG221* and downstream of the 70 bp repeats in the active BES (BES1), fused to a *PUR* gene (*Glover et al., 2013*), an organisation similar to that described by *Boothroyd et al. (2009)*. Here, DSB induction has been proposed to mimic VSG switching, by initiating HR through available homology (e.g. other *VSG*s, 70 bp repeats, telomere repeats, *ESAG*s)(*Boothroyd et al., 2009*; *Glover et al., 2013*). In both HR1 and HRES, the presence or absence of the *PUR* gene at the I-SceI recognition site provides a means to assay for repair after Tet induction: due to the proximity of the *PUR* gene to the I-SceI target, the *PUR* sequence must be degraded after I-SceI cutting by DSB end resection to access the flanking homology that drives HR-directed DSB repair DNA, resulting in puromycin sensitivity.

TbRECQ2 mutants were generated in HR1 and HRES BSF cells as before, with integration of the constructs and loss of intact *RECQ2* in two -/- mutant clones confirmed by PCR (*Figure 3— figure supplement 1A*) and RT-PCR (*Figure 3—figure supplement 1B*). In order to understand if loss of TbRECQ2 affected DSB repair, cell survival following Tet induction was assayed by determining clonal survival efficiency relative to uninduced cells. The survival rate of HR1 WT cells following I-SceI induction was ~60% (*Figure 3B*), equivalent to that reported previously (*Glover and Horn, 2014*; *Glover et al., 2008*). In the absence of TbRECQ2, the survival rate decreased 2-fold, with only ~30% of wells displaying growth in both of the two HR1 *recq2-/-* clones examined. These data indicate that *recq2-/-*mutants are less able to survive a chromosome-internal DNA DSB than WT *T. brucei* cells. Analysis of the puromycin sensitivity of surviving clones showed, for both WT and *recq2-/-* HR1 cells, that all Tet-induced clones (n = 25) were puromycin sensitive and all uninduced clones (n = 6) were puromycin resistant (*Figure 3B*). Thus, a functional *PUR* gene was lost in all cases after I-SceI expression was induced, showing that DSB formation was successful and indicating repair is possible, though less efficient, in the absence of TbRECQ2. Broadly the same outcome was seen for the HRES cells (*Figure 3C*). Consistent with previous observations (*Glover et al., 2013*), the survival rate of HRES WT cells after I-SceI induction (~24%) was >2-fold lower than HR1 cells (*Figure 3C*), indicating greater lethality when a DSB is made in the active BES. Nonetheless, the survival rate of the two *HRES recq2-/-* clones (13% and 14.8%) was again ~50% of HRES WT cells, suggesting loss of TbRECQ2 impairs survival in both loci. In fact, the level of impairment after loss of TbRECQ2 may be greater than the clonal survival assay predicts since, unlike in HR1, evaluating the puromycin sensitivity of recovered clones showed that 14% of survivors (n = 7) in one *HRES recq2-/-* clone and 70% (n = 10) in the other were puromycin resistant (*Figure 3C*). These data are most simply explained by a greater number of HRES *recq2-/-* mutants being recovered (relative to HR1 mutants) in which a DSB has not been induced, reflecting the very limited survival capacity of

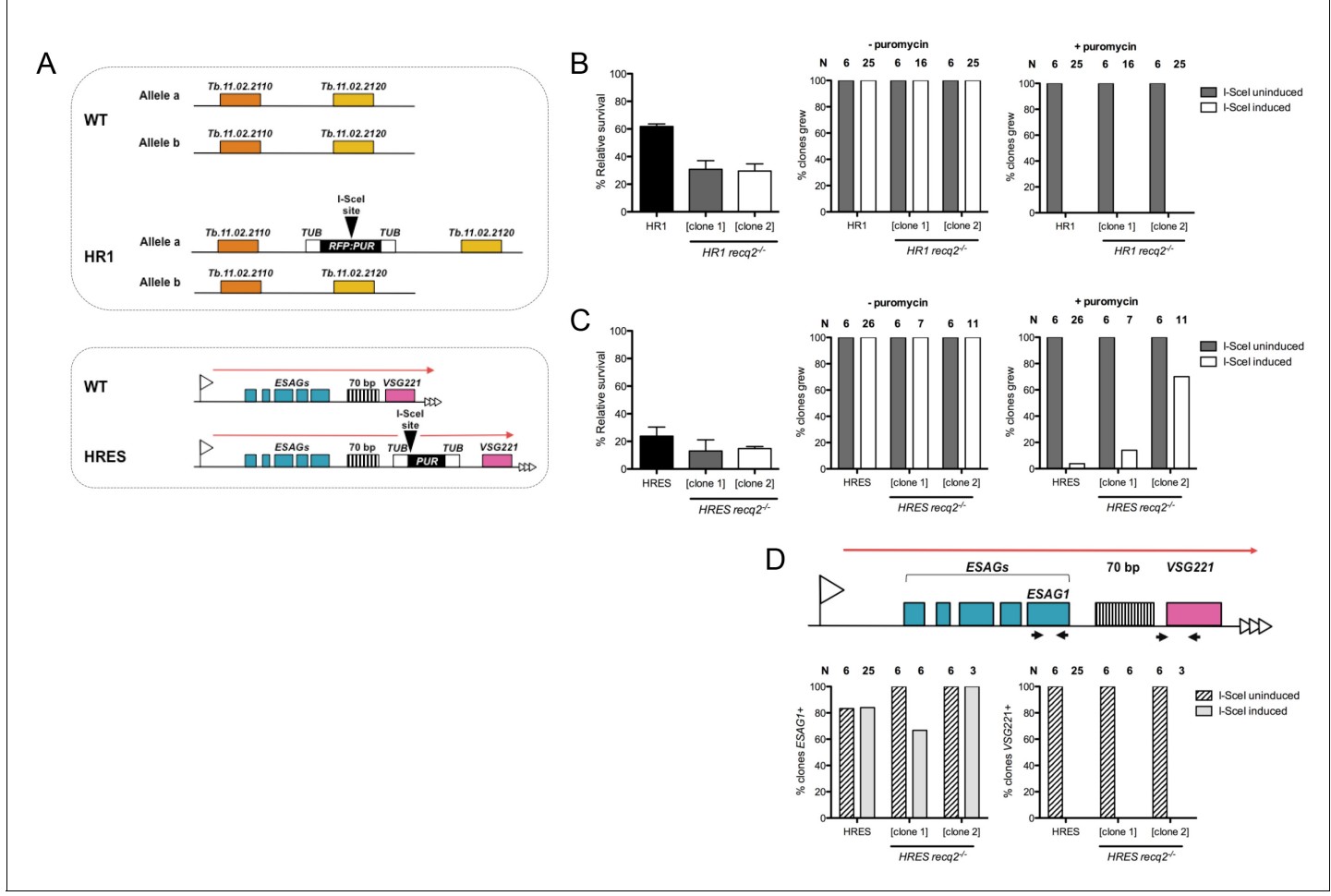

**Figure 3.** Mutation of *TbRECQ2* impairs survival of *T. brucei* after induction of a DNA double strand break, either in the active telomeric VSG expression site or in the core of a chromosome. (**A**) I-SceI target sequences in HR1 and HRES cells. HR1 cells contain an I-SceI recognition site embedded within an *RFP:PUR* fusion gene (black), flanked by tubulin sequences (white), located between genes Tb.11.02.2110 and Tb.11.02.2020 on one copy of chromosome 11; HRES cells contain an I-SceI recognition site upstream of a *PUR* gene, flanked by tubulin sequences, located downstream of the 70 bp repeats of the active *VSG221* expression site on chromosome 6. **B** and **C** show clonal survival following I-SceI induction in HR1 and HRES cells, respectively. In both cases, wild type and two *recq2-/-* clones were distributed in three 96 well plates at a concentration of 0.26 cells per well either in the absence (I-SceI uninduced) or presence (I-SceI induced) of 2 μg.mL$^{-1}$ tetracycline. The number of wells with surviving cells after 7–10 days growth is depicted as percentage of survivors following I-SceI induction relative to survivors without I-SceI induction; error bars represent standard error of the mean between three experimental repeats. Puromycin sensitivity of surviving I-SceI induced and uninduced clones was then tested, and is represented as the percentage of tested clones that grew in the presence (+) or absence (-) of 1 μg.mL$^{-1}$ puromycin (N: number of clones analysed). (**D**) Clones from (**C**), excluding those that were puromycin resistant, were assayed for *ESAG1* and *VSG221* presence by PCR; data are shown as the percentage that were PCR positive (N: number of clones analysed).

The following figure supplement is available for figure 3:

**Figure supplement 1.** Generation of *TbRECQ2* mutants in *T. brucei* HR1 and HRES cells.

*recq2-/-* mutants after a DSB is made in the active BES. Despite this, PCR of the puromycin sensitive *recq2-/-* survivor clones showed that all had lost the *VSG221* gene (also called *VSG 427–2*) (*Hertz-Fowler et al., 2008*) and most had retained the *ESAG1* gene variant specific to the targeted BES (Figure 3D), a pattern consistent with previous analysis in WT HRES (*Glover et al., 2013*) cells and relatives (*Boothroyd et al., 2009*). Thus, loss of TbRECQ2 does not result in a major shift in repair pathway after induction of a DNA DSB in the BES, meaning reduced survival in the mutants is best explained by less efficient execution of a predominant repair reaction.

## Loss of TbRECQ2 increases VSG switching by elevated telomeric recombination

In order to analyse the effect of TbRECQ2 loss onVSG switching we adapted an in vitro strategy of Povelones et al (*Povelones et al., 2012*), which is conceptually related to an assay established by Kim and Cross (*Kim and Cross, 2010; 2011*). In this strategy, a herpes simplex virus thymidine kinase (*TK*) gene fused to a hygromycin resistance gene (*HYG-TK*) was inserted between the 70 bp repeats and *VSG221* in the active BES (*Figure 4A*). Additionally, enhanced *GFP* and *PUR* genes were integrated downstream of the active BES promoter. Integration of the marker genes and the expected expression of GFP and VSG221 protein was confirmed by PCR and western blotting (*Figure 4—figure supplement 1A*).The parental cell line generated by these manipulations, *GFP221hygTK*, allows the nucleotide analogue ganciclovir (GCV) to be used to eliminate cells from the population that have not inactivated *TK*, which can occur through VSG switching events that can be distinguished by the presence and expression of the *VSG221* and *GFP* genes in the BES (*Figure 4A*). In a transcriptional (in situ) switch VSG221 and GFP proteins are no longer expressed from the BES but both genes are retained. In contrast, cells that have switched by a gene conversion downstream of *PUR-GFP* (here termed VSG GC) retain GFP expression from the BES but have deleted *TK* and *VSG221* from the transcription unit. Longer range gene conversions are also possible that encompass the whole BES and lead to removal of both *GFP* and *VSG221*, though in the approach used here this reaction cannot be distinguished from events in which the BES is deleted without gene conversion and cells survive through a transcriptional switch (ES GC or in situ+ES del, respectively)(*Cross et al., 1998*; *Rudenko et al., 1998*). Finally, in VSG switching by telomere exchange (telomere XO) GFP continues to be expressed from the BES, whereas VSG221 protein expression is silenced by moving the gene to another telomere. Cells that have inactivated *TK* through mutation, rather than VSG switching, can also be selected for in this assay (*Povelones et al., 2012*). However, such cells, which can be identified by continued expression of VSG221, were rare in this study (*Figure 4C*).

To assess the contribution of TbRECQ2 to VSG switching, *req2+/-* and *req2-/-* mutants were generated in the *GFP221hygTK* cells, with integration of the constructs and loss of intact *TbRECQ2* in the -/- mutants confirmed by PCR, and continued expression of GFP and VSG221 shown by western blotting (*Figure 4—figure supplement 1B*). The switching rate of the WT cells was then compared with the *TbRECQ2* mutants: cultures were grown for 48 hr in media lacking hygromycin or puromycin (allowing switch variants to arise) and then cloned by limiting dilution in antibiotic-free media containing GCV, allowing measurement of the number of cells in the diluted population that had inactivated *TK*. In the WT cells GCV resistant clones arose at a rate of ~1.5 x $10^{-5}$cells/generation (*Figure 4B*), consistent with rates determined in comparable studies Kim and Cross, 2010a, 2011a; Povelones et al., 2012b). No change in rate was seen in the *recq2+/-* mutants (*Figure 4B*), but GCV resistant cells arose around 2-fold more frequently in the *recq2-/-* mutants (~3.2 x $10^{-5}$ resistant cells/generation). To ask if this change could be explained by an alteration in VSG switching strategy, we used PCR and western blotting to determine the profile of *GFP* and *VSG* gene presence and protein expression in a selection of GCV resistant clones for each cell line (*Figure 4C*; *Figure 4—figure supplement 2*). In broad agreement with observations made by *Povelones et al. (2012)*, in WT cells most GCV resistant clones had arisen either by in ES GC or in situ+ES del (~60%), with in situ switching the next most common process (~25%); VSG GC was rare (~10%), and we found no examples of telomere XO events. A virtually identical pattern of events was seen in the *recq2+/-* cells, consistent with the unaltered rate at which GCV resistant cells arose. In contrast, ~90% of GCV resistant clones in the *recq2-/-* mutants arose either by VSG GC (~50% of total) or telomere XO (~45%), indicating that the elevated rate in the null mutants is due to increased use of these recombination strategies. To test this interpretation further, the switching experiment was conducted in the presence of puromycin, which should prevent any events that inactivate expression of *PUR* (*Figure 4B*). In these conditions, GCV resistant cells arose ~2–3 fold less frequently in the WT and *recq2+/-* cells (*Figure 4C*), consistent with their predominant use of in situ and ES GC or in situ+ES del events, whilst there was less impact on the *recq2-/-* cells, where VSG switching is largely downstream of *PUR*. Taken together, these data indicate that loss of TbRECQ2 results in increased VSG switching by a change in repair strategy.

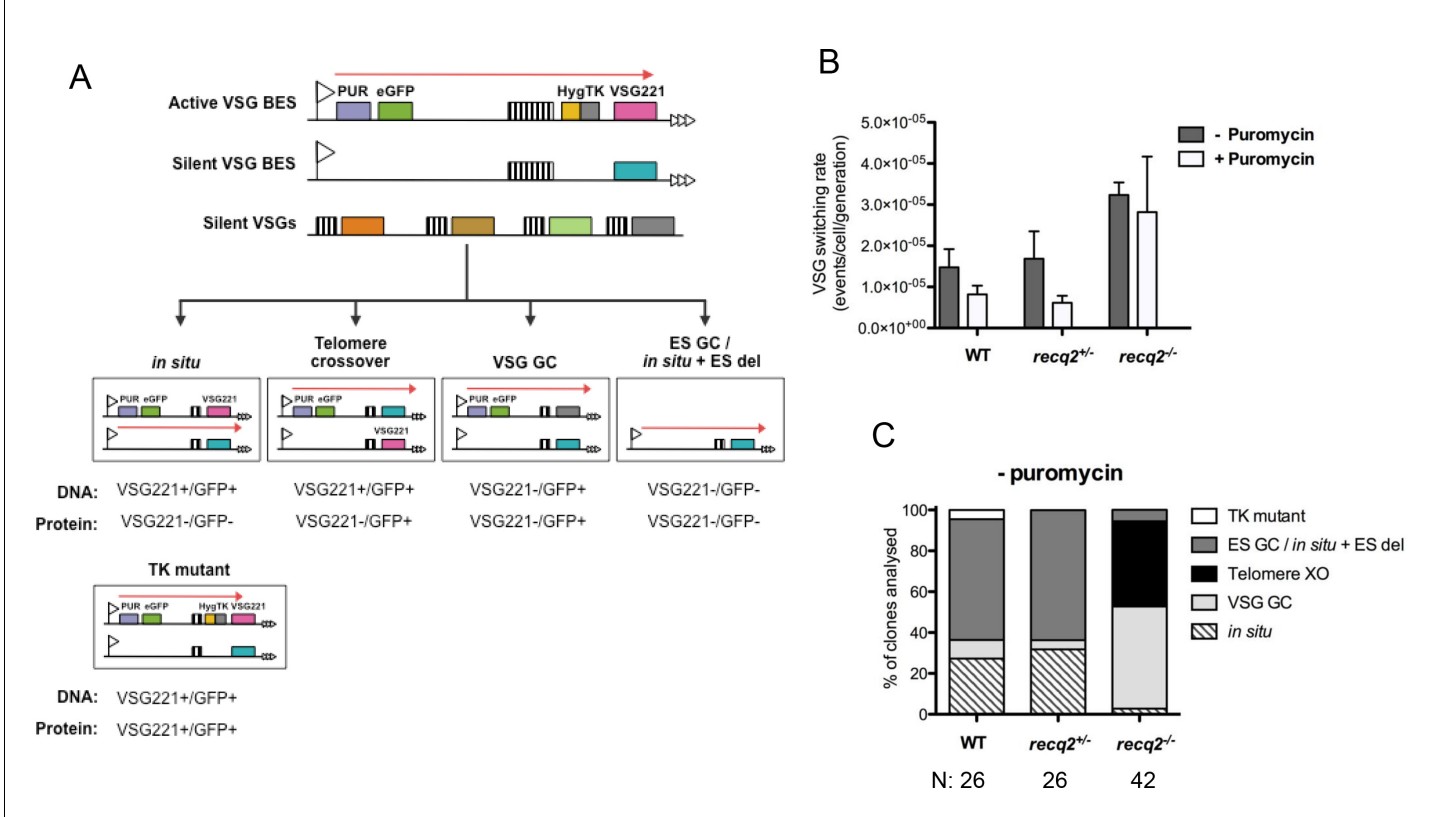

**Figure 4.** Mutation of TbRECQ2 leads to elevated VSG switching and increased recombination. (**A**) Strategy for determining VSG switching mechanisms; adapted from *Povelones et al. (2012*). The active *VSG* BES of *GFP221hygTK* cells is shown, within which the *PUR*, e*GFP, HYG-TK* and *VSG221* genes are represented as coloured boxes. In addition, one of the ~14 silent BES containing a distinct *VSG* (turquoise box) is shown, as are multiple silent *VSG*s elsewhere in the genome (various colours; for convenience these are shown as a single array, but could also be at the telomere of silent mini-chromosomes). 70 bp repeats upstream of the *VSG*s are denoted by hatched boxes. Different switching strategies allow survival after ganciclovir treatment and can be distinguished by analysis of *VSG221* and *GFP* presence by PCR, and expression of the proteins by western blot (profiles detailed under each mechanism). Switchers that arise by in situ switching, telomere crossover (XO) or VSG gene conversion (VSG GC) can be detected unambiguously, while events that occur by BES gene conversion or in situ switching coupled with BES deletion (ES GC/ in situ+ES del) are indistinguishable. Note, only in situ+ES del reaction is shown, and not ES GC (where all sequence of a silent BES is duplicated and replaces the *VSG221* BES); in addition, for VSG GC the silent grey array donor *VSG* gene is shown as being copied, but the reaction could also use a BES *VSG* gene. Non-switcher TK mutants can also allow ganciclovir survival. (**B**) The mean switching rate of *GFP221hygTK* WT and *recq2* mutants (+/- and -/-) was inferred from the mean number of survivors from two experiments, each with three replicates, following treatment with ganciclovir and after culture with (+) or without (-) puromycin; error bars represent standard error of the mean. (**C**) Profiles of WT and *recq2* mutants (+/- and -/-) survivors in the non-puromycin experiments, represented as a percentage of total surviving clones analysed from the two datasets; number of clones (N) analysed is indicated.

The following figure supplements are available for figure 4:

**Figure supplement 1.** Generation of *TbRECQ2* mutants in *T. brucei GFP221hygTK* cells.

**Figure supplement 2.** Summary table of ganciclovir survival mechanisms.

## A DNA double strand break in the VSG expression site is inefficiently repaired

Given the dichotomy between the effects of TbRECQ2 loss on DNA DSB repair and VSG switching, we next characterised in more detail the response of *T. brucei* BSF cells to induction of an I-SceI-mediated DSB. We first compared the cell cycle response to DSB induction in HR1 and HRES cells (*Figure 5A,B*). Expression of I-SceI was induced with Tet and, 12 and 24 hrlater, cells were stained with DAPI to visualize N and K DNA. Virtually all cells were categorised as either 1N1K, 1N2K or 2N2K, irrespective of whether I-SceI expression was induced or not. However, consistent with

previous reports (*Glover et al., 2013*; *Glover and Horn, 2014*; *Glover et al., 2008*), addition of Tet resulted in increased numbers of 1N2K cells, indicating impaired G2-M cell cycle progression (*Figure 5A,B*). In HR1, 1N2K cell numbers increased (from ~10% of the uninduced population) to ~20% 12 hr after Tet induction and then returned to ~10% after 24 hr (*Figure 5A*), indicating the cell cycle impairment was transient. In contrast, increased 1N2K cell numbers (~20% of the population) persisted until 24 hr after Tet addition in HRES (*Figure 5B*), suggesting the response to a DSB differs if the lesion is in the active BES or a chromosome-internal site (*Glover and Horn, 2014*). To examine this further, we used quantitative real-time PCR (qPCR) to assess the dynamics of I-SceI site cleavage.

Genomic DNA was prepared at multiple timepoints after Tet induction in HR1 or HRES cells and qPCR was performed with primers spanning the I-SceI target sequence, determining the amount of PCR product after Tet induction relative to uninduced cells and normalized by a control locus (ΔΔCt

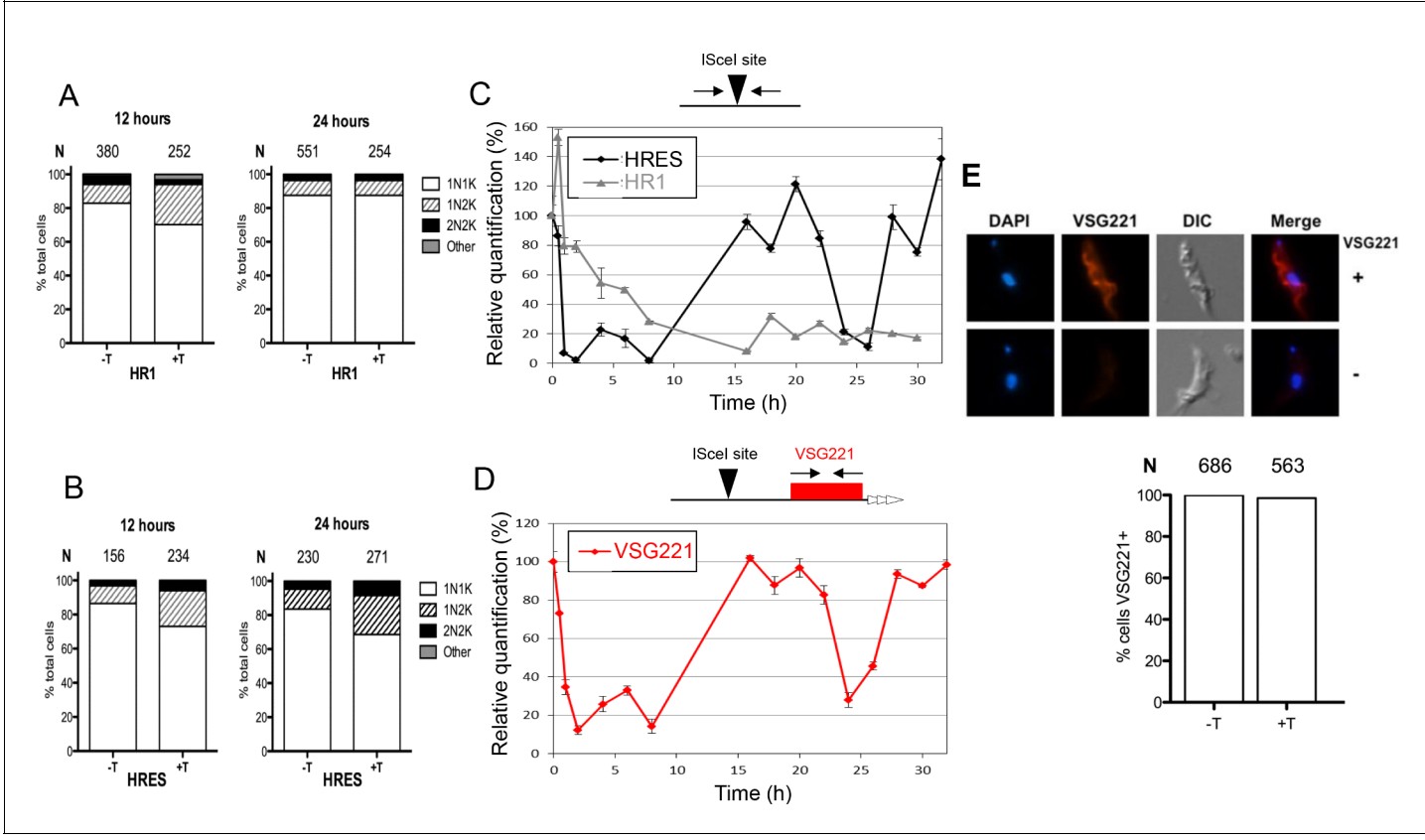

**Figure 5.** Analysis of cell cycle progression, DNA repair kinetics and VSG expression after I-SceI-mediated DNA double strand break formation. Cell cycle analysis of HRES (**A**) and HR1 (**B**) cells following I-SceI induction. DNA content is shown 12 and 24 hr post I-SceI induction (+T) after visualisation by DAPI staining of fixed cells; uninduced cells (-T) were analysed as a control. The number of cells with one nucleus and one kinetoplast (1N1K), one nucleus and two kinetoplasts (1N2K), two nuclei and two kinetoplasts (2N2K) and cells that did not fit into any of these categories (other) were counted. The proportion of each cell type is represented as a percentage of the total cells counted (**N**). (**C**) Relative efficiency of PCR amplification of the I-SceI target sequence is shown at various time points after induction of I-SceI in HR1 and HRES cells; values are shown at each post-induction time point as a percentage of the amount of PCR product generated at 0 hr; values are the mean of three experimental repeats and vertical lines denote standard deviation. (**D**) Relative PCR amplification of the *VSG221* gene downstream of the I-SceI target is shown after I-SceI induction in HRES cells; values were determined and are represented as in **C**. (**E**) VSG221 expression in HRES cells 24 hr post I-SceI induction was visualised by indirect immunofluorescence of fixed cells with anti-VSG221 antiserum (+), and are compared with control cells in which only secondary antiserum was used (-). The graphs below show the proportion of cells expressing VSG221 on their surface 24 hr post I-SceI induction (+T), or without I-SceI induction (-T); data are represented as the percentage of total cells counted (**N**).

The following figure supplement is available for figure 5:

**Figure supplement 1.** Break formation over 72 hr after ISceI induction in *T. brucei* HR1 cells.

method))(*Livak and Schmittgen, 2001*). In HR1 (*Figure 5C*), a reduction in product was seen from 2 hr after Tet addition (20%) and this effect increased until around 8 hr (80%), with little change in PCR efficiency thereafter (8–72 hr; *Figure 5C* and *Figure 5—figure supplement 1*). These data suggest that I-SceI cleavage in HR1 cells is rapid and reaches a maximum within a cell cycle (~8 hr), an interpretation consistent with the timing of single-stranded DNA formation at I-SceI breaks detected by Southern blotting (*Glover and Horn, 2014*; *Glover et al., 2008*). Southern blotting suggests that allelic repair products in HR1 accumulate slowly over ~16–72 hr (*Glover et al., 2008*), consistent with qPCR to detect duplication of the intact allele (data not shown). I-SceI target qPCR revealed a different response to I-SceI induction in HRES cells (*Figure 5C*). Early in the time course, loss of PCR product was more rapid and more complete, with <10% of uninduced product by 2–3 hr after Tet addition, suggesting I-SceI cleavage was at least as efficient in HRES cells as in HR1. However, in contrast with HR1, the levels of PCR product later in the reaction increased, in two distinct peaks (at ~20 and 30 hr post induction), to levels approaching that of the uninduced cells. Given that I-SceI cleavage is rapid, these data either suggest repair occurs efficiently in a manner that regenerates the I-SceI target, or repair is inefficient and cells that are subjected to I-SceI cleavage are killed, allowing those that have not suffered a DNA DSB to outgrow. Unlike for HR1, where most repair is mediated by allelic HR, I-SceI cleavage in HRES (or related cells) predominantly results in cells expressing a new *VSG*, whose identity cannot be easily predicted (*Figure 3*)(*Boothroyd et al., 2009*; *Glover et al., 2013*). Thus, we used qPCR to ask about the fate of the BES *VSG (VSG221)* after I-SceI cleavage (*Figure 5D*). The abundance of the *VSG221* PCR product very closely matched that of the I-SceI target over the timecourse, and the recurrence of product indicates the initial, rapid loss of *VSG221* is not due to replacement by another *VSG*. To test this further, we performed immunofluorescence with anti-VSG221 antiserum, revealing that virtually all cells retained VSG221 on their surface 24 hr after Tet induction (*Figure 5E*). Taken together, these data indicate that induction of a DSB by I-SceI in the active BES does not elicit rapid repair (within 32 hr) that removes the downstream *VSG*.

## MFAseq reveals early replication of the single actively transcribed *VSG* expression site in mammal-infective *T. brucei*

The analyses above question the association between an induced DNA DSB and VSG switch initiation. In order to ask if any other feature of genome maintenance might correlate with antigenic variation, we examined the dynamics of *T. brucei* nuclear DNA replication using marker frequency analysis coupled with next generation sequencing (MFAseq) (*Tiengwe et al., 2012*). MFAseq compares the relative depth of sequence read mapping in replicating (S phase) and non-replicating (here, G2) cells, allowing the sites and relative efficiencies of origins of replication to be determined, as well as inference on the timing and direction of replication genome-wide. In *T. brucei*, MFAseq has so far only been performed in procyclic cell forms (PCF), the insect stage of the parasite, and in the strain TREU927 (*Tiengwe et al., 2012*), in which the repertoire of telomeric BES has not been characterized. Here, we performed MFAseq in PCF and BSF cells of *T. brucei* strain Lister 427, where all BES have been sequenced (*Hertz-Fowler et al., 2008*), allowing us to ask if differences between the life cycle stages, including gene expression changes, result in alterations in replication dynamics. *Figure 6* shows MFAseq mapping for the eleven megabase chromosomes of *T. brucei*, excluding the BES, and comparing the patterns seen when early S or late S cells are compared with G2 phase cells. The MFAseq pattern of peak location was invariant when comparing early S BSF and PCF cells, and when comparing late S phase BSF and PCF cells. Even more strikingly, the relative heights of the MFAseq peaks in each chromosome were invariant between the two life cycle stages (*Figure 6*), as well as being invariant between *T. brucei* strains Lister427 and TREU927 (data not shown). Thus, neither differentiation between life cycle stages in *T. brucei*, nor extended growth of different *T. brucei* strains, leads to changes in the genomic sites used as origins, or changes in the timing programme of origin activation in the chromosome cores. Late S MFAseq has not previously been reported for *T. brucei*, though it was inferred that the number of origins (42) mapped using early-mid S phase cells might be an underestimate of ~2-fold, due to replication initiation at late acting origins (*Tiengwe et al., 2012*). This prediction appears to be inaccurate, as most of the peaks detected in the late S samples were merely wider than the early S peaks; indeed, in several locations early S peaks had merged as replication forks converged (*Figure 6*). Only five origins (dashed lines, *Figure 6*) were observed in the present data, both in the BSF and PCF cells, which were not

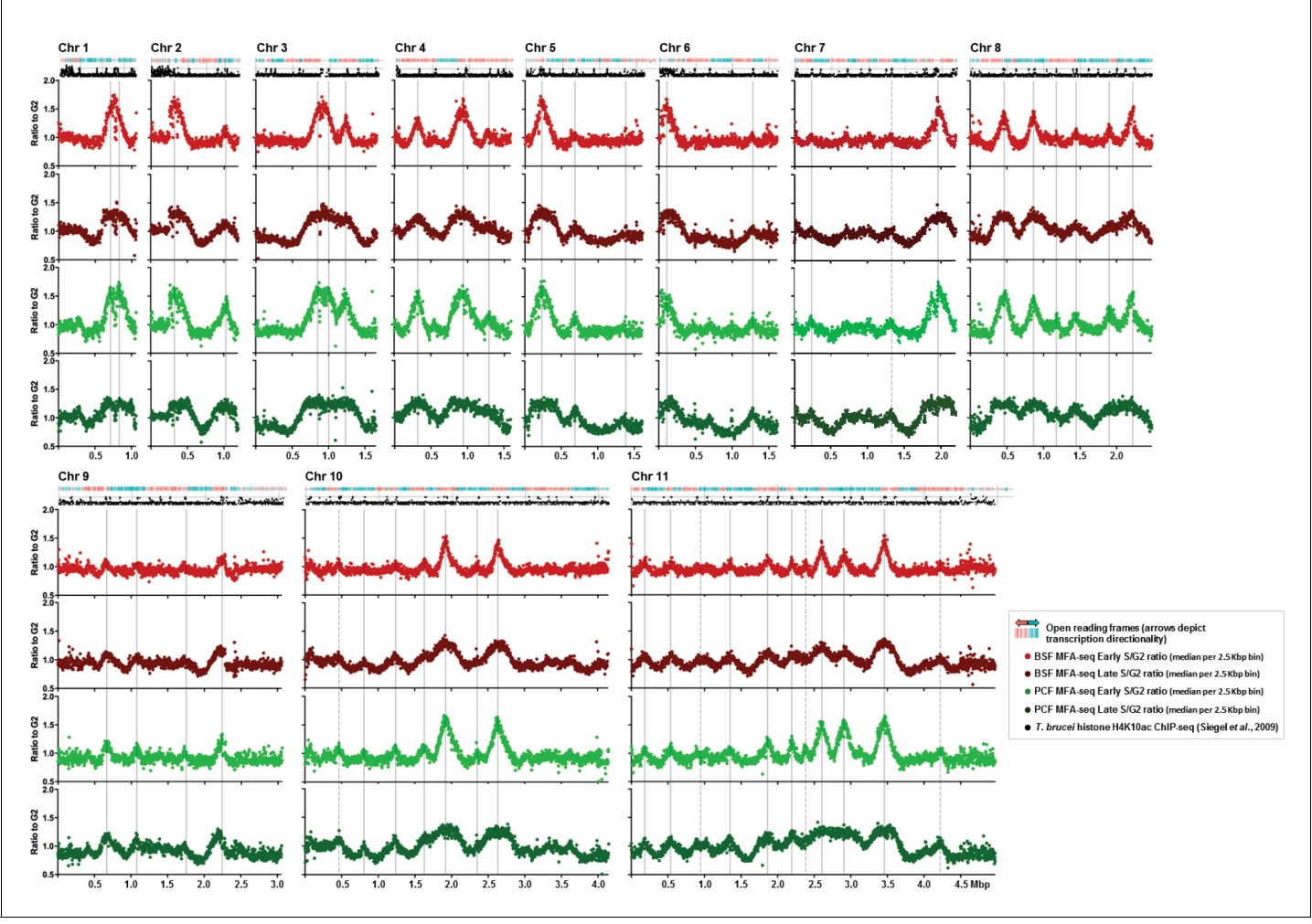

**Figure 6.** Replication timing throughout the core genome is stringently conserved between BSF and PCF *T. brucei* cells. Each set of four graphs shows the distribution of replication origins in the 11 megabase chromosomes (depicted as Chr1 to Chr11), assessed by MFAseq (*Tiengwe et al., 2012*). At the top of each set of graphs is a track representing the genes in the chromosome: in blue the open reading frames (ORFs) are transcribed from the left to the right, and in red they are transcribed from right to left. Below each set of graphs is a track depicting histone H4K10ac-enriched sites (*Siegel et al., 2009*). The four graphs in each case show the ratio between the coverage (read-depth) of DNA derived from Illumina sequencing of early S phase and G2 phase cells, or late S phase and G2 phase cells, where each point represents the median S/G2 ratio (y-axis) in 2.5 Kbp bins across the chromosome (x-axis; bars indicate 500 Kbp intervals). All graphs are scaled according to chromosome size. The light red graph shows MFAseq for BSF early S cells, while dark red represents the data from late S phase. PCF MFAseq data is shown for early S cells in light green, and in dark green for late S. Vertical, solid grey lines represent the origins identified previously (*Tiengwe et al., 2012*), while dashed lines highlight replication origins only observed in this study.

predicted previously (*Tiengwe et al., 2012*). All these were 'weak' origins, with low MFAseq peak heights, and it is possible that they were observed here due to the more compressed graphical representation, rather than being origins that are active in Lister 427 cells and not in TREU927. Using the localisation of TbORC1/CDC6 (*Tiengwe et al., 2012*) and histone H4K10Ac (*Siegel et al., 2009*) binding sites in the TREU927 genome as a guide (*Figure 6*), it is clear these five origins localise to the boundaries of the polycistronic transcription units, as expected. Taken as a whole, the above MFAseq analysis suggests pronounced rigidity in the coordination of nuclear DNA replication in *T. brucei*.

The BES repertoire of *T. brucei* Lister 427 is composed of 14 distinct BES (*Hertz-Fowler et al., 2008*), most of which have not been allocated to specific megabase or intermediate chromosomes. The PCF and BSF S and G2 sequence reads were used to perform MFAseq mapping to the 16 available contigs representing the 14 different BES, as shown in *Figure 7*. In these data, peaks cannot

be discerned, as the sizes of the BESs are smaller than the distance covered by the replication forks at most origins (*Figure 6*). Thus, at the higher resolution used here, the MFAseq mapping is seen as multiple discrete points, corresponding to the median S/G2 read depth ratio in each of the 2.5 kbp 'bins' that span the BES (*Figure 7B*). For all but one of the BES, there was no clear difference between the MFAseq mapping in the BSF and PCF cells, either for early or late S phase. Moreover, there was no evidence that 13 of the 14 BES had been replicated, even in the BSF and PCF late S samples, since there was no consistent increase in S phase reads relative to G2: comparing the S/G2

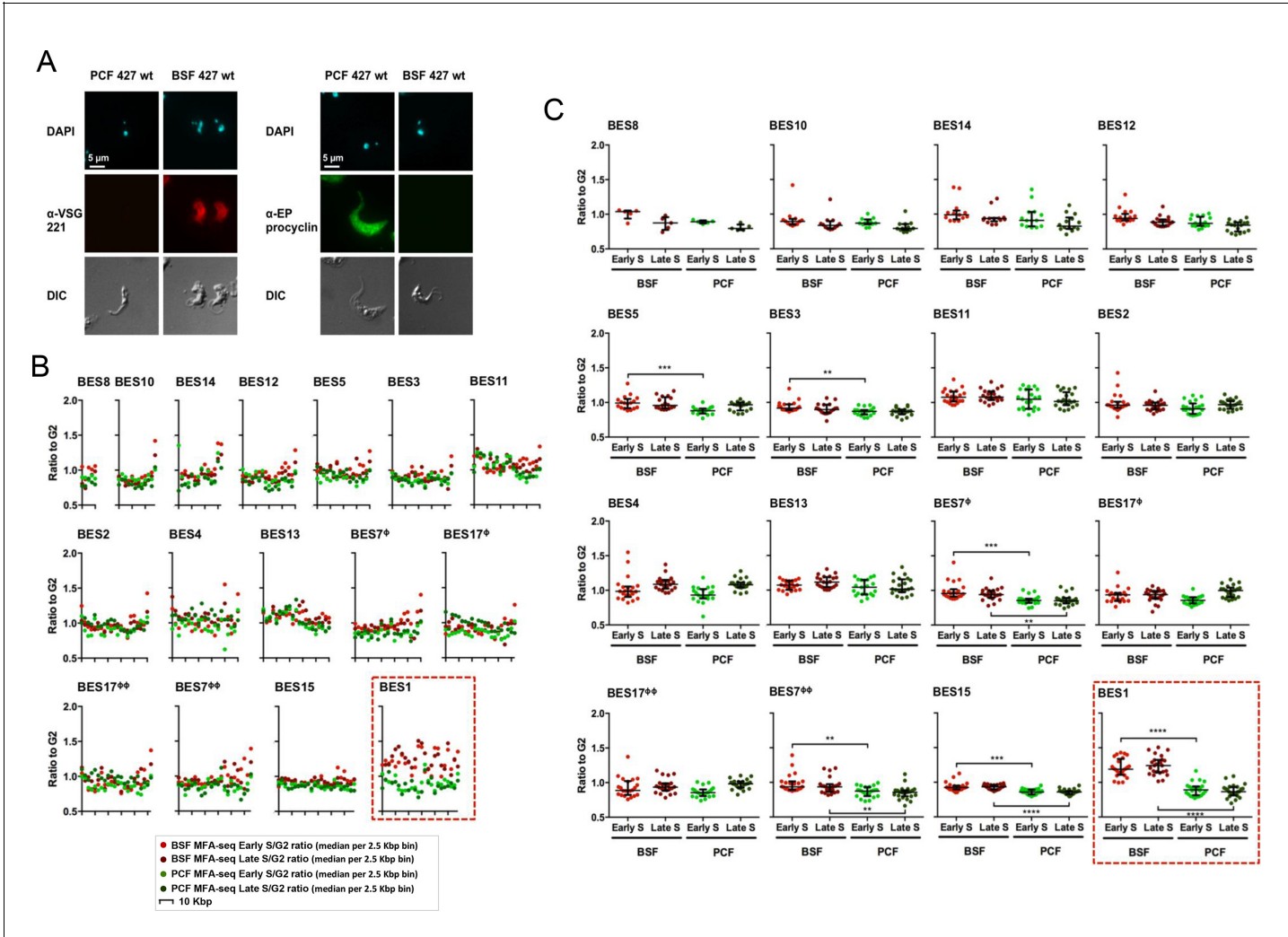

**Figure 7.** The active VSG expression site in bloodstream form *T. brucei* cells is the only telomeric site that is early replicating. (A) Immunofluorescence of PCF or BSF Lister 427 cells and BSF strain Lister 427 with anti-VSG 221 antiserum or with anti-EP procyclin antiserum; top panels show the cells stained with DAPI, while the bottom panel shows the cells' outline by DIC. Images were acquired with the Axioskop 2 imaging system and the scale bar represents 5 μm. (B) The Lister 427 bloodstream VSG expression site (BES) TAR clones sequenced by (Hertz-Fowler et al., 2008) were used to map the MFAseq data from BSF and PCF cells; note that two BES are represented by duplicate TAR clones: BES 7 (φ – TAR 65; φφ – TAR 153), and BES 17 (φ – TAR 51; φφ – TAR 59). The ratio between sequence coverage (read-depth) in early S phase and G2 phase cells, or late S phase and G2 phase samples, is plotted, where each point represents the median S/G2 ratio (y-axis) per 2.5 Kbp bin across the BES (x-axis). The size of each BES is shown on each x-axis in 10 Kbp intervals, and all graphs are scaled according to BES size. The y-axis scale is the same for all graphs, but the legend is only shown on the ones at the far left. BSF early S data is represented as light red, BSF late S as dark red, PCF early S as light green, and PCF late S as dark green. The red dashed box highlights BES 1. (C) The S/G2 values used to generate the graphs in (B) are shown plotted per sample (BSF early S – light red, BSF late S – dark red, PCF early S – light green, and PCF late S – dark green), rather than by genomic location, for each BES (numbered as before). Horizontal bars (black) represent the median of the S/G2 values, and error bars the interquartile range. In order to infer statistical significance, the values were analysed with the non-parametric, unmatched, Kruskal-Wallis test; statistical significance is only shown for differences between the BSF and PCF samples: (**) p-value <0.01; (***) p-value <0.001; (****) p-value <0.0001.

ratios for each bin in each BES showed that the overall median S/G2 ration for most BES was ~1.0 (*Figure 7C*). These data suggest that 13 of 14 BES are replicated very late in S phase, similar to telomeres in other eukaryotes (*Rhind and Gilbert, 2013*). BES1 was the single exception to the above trend. In this contig, BSF S phase reads (both early and late) across the BES were markedly elevated relative to G2 (*Figure 7B*), with overall median S/G2 ratios of 1.25 and 1.3 in early and late S, respectively (*Figure 7C*). This effect was limited to the BSF cells, however, since the PCF MFAseq data for BES1 (again in either early or late S) was comparable with all other BES, with S/G2 ratios ~1.0 (significantly different from BSF cells: p-value < 0.0001). These data suggest that BES1, alone amongst the 14 BES, is replicated early, and this deviation from the late replication of other BES is limited to mammal-infective cells. BES1 differs from the other BES in being the actively transcribed site, encoding VSG221 in the BSF cells used in this study, as shown by indirect immunofluorescence (*Figure 7A*). In PCF cells full transcription of all BES, including BES1, is silenced (*Rudenko et al., 1994*) and the VSG coat is replaced with procyclin (*Figure 7A*). Thus, the unique early replication of BES1 only in BSF cells suggests that replication timing of *T. brucei* telomeres displays a precise association with transcription.

To test the MFAseq mapping in the BES we used qPCR to examine the predicted early replication of the actively transcribed *VSG* gene. First, S/G2 ratios were derived using qPCR on DNA from cells expressing BES1 (*VSG221*), which had been used in the MFAseq (*Figure 6* and *7*). Using an origin-associated locus and a non-origin locus from chromosome 5 (*Figure 8A*) as controls, we observed early and late S/G2 qPCR ratios consistent with the former locus being early replicating and the latter late replicating (*Figure 8B*). Furthermore, qPCR of *VSG221* in the same cells revealed early and late S/G2 ratios higher than the chromosome 5 origin locus, consistent with the *VSG* replicating very early (*Figure 8B*). To ask if the early replication of *VSG221* is determined by transcription of BES1, we next examined cells generated by Glover et al (*Glover et al., 2007*) in which a Tet operator is inserted downstream of the BES1 promoter (*Figure 8B*). In the absence of Tet, binding of the Tet repressor (TetR) blocks transcription elongation in BES1 and cells are selected that have switched to transcribing another *VSG* BES (*Aresta-Branco et al., 2016*; *Glover et al., 2007*). A recently derived clone of the BES1 TetR blockade cells was sorted into early S, late S and G2 populations, and qPCR performed on recovered DNA. Early and late S/G2 qPCR ratios for the chromosome 5 controls were lower in these experiments compared with the qPCR from the BES1 (VSG221) expressers, probably as a result of sorting differences in the selection of cells within the S phases or within G2. Nonetheless, the S/G2 ratio of the origin locus increased from early to late S, indicating replication progression. For *VSG221*, the early and late S/G2 ratios in the TetR blockade cells were much lower than in the BES1 (*VSG221*) expressing cells, being indistinguishable from the origin locus, indicating the pronounced early replication of this *VSG* is not seen when it is no longer transcribed. Immunofluorescence indicated that VSG121, whose gene is present in BES3 in this *T. brucei* strain (*Hertz-Fowler et al., 2008*), could be detected on the cell surface of most of the BES1 TetR blockade cells at the outset of the experiment (*Figure 8B*) and so we used qPCR to test the replication timing of this *VSG* in both cell types. qPCR of *VSG121* is complicated relative to *VSG221* because the gene is not only located in a BES: at least four *VSG121* copies are found within the subtelomeric *VSG* arrays (*Trenaman et al., 2013*), whose replication timing is unclear (*Tiengwe et al., 2012*). Despite this, early and late S/G2 qPCR ratios for *VSG121* in BES1 (*VSG221*) expressing cells were lower than both the *VSG221* and chromosome 5 origin control locus values and were more comparable with the non-origin control (*Figure 8B*), consistent with late replication. In contrast, in the BES1 TetR blockade cells the *VSG121* S/G2 ratios were higher than both *VSG221* and the non-origin locus and, instead, were comparable with the origin control (*Figure 8B*). Thus, despite the potentially confounding effect of *VSG121* array copies that may be late replicating, as well as uncertainty about the transcriptional status of all the VSG BES in these cells, these data suggest earlier replication of *VSG121* when the *VSG221*-containing BES1 is silenced and BES3 is at least one of the BES expressed in the *T. brucei* population. Taken together, these *VSG*-focused qPCR experiments validate the MFAseq association between replication timing and transcription status of the telomeric BES.

## Discussion

Understanding the initiation event(s) of VSG switching by recombination is important, since this element of the reaction may be lineage-specific, and might explain both the elevated rate of the

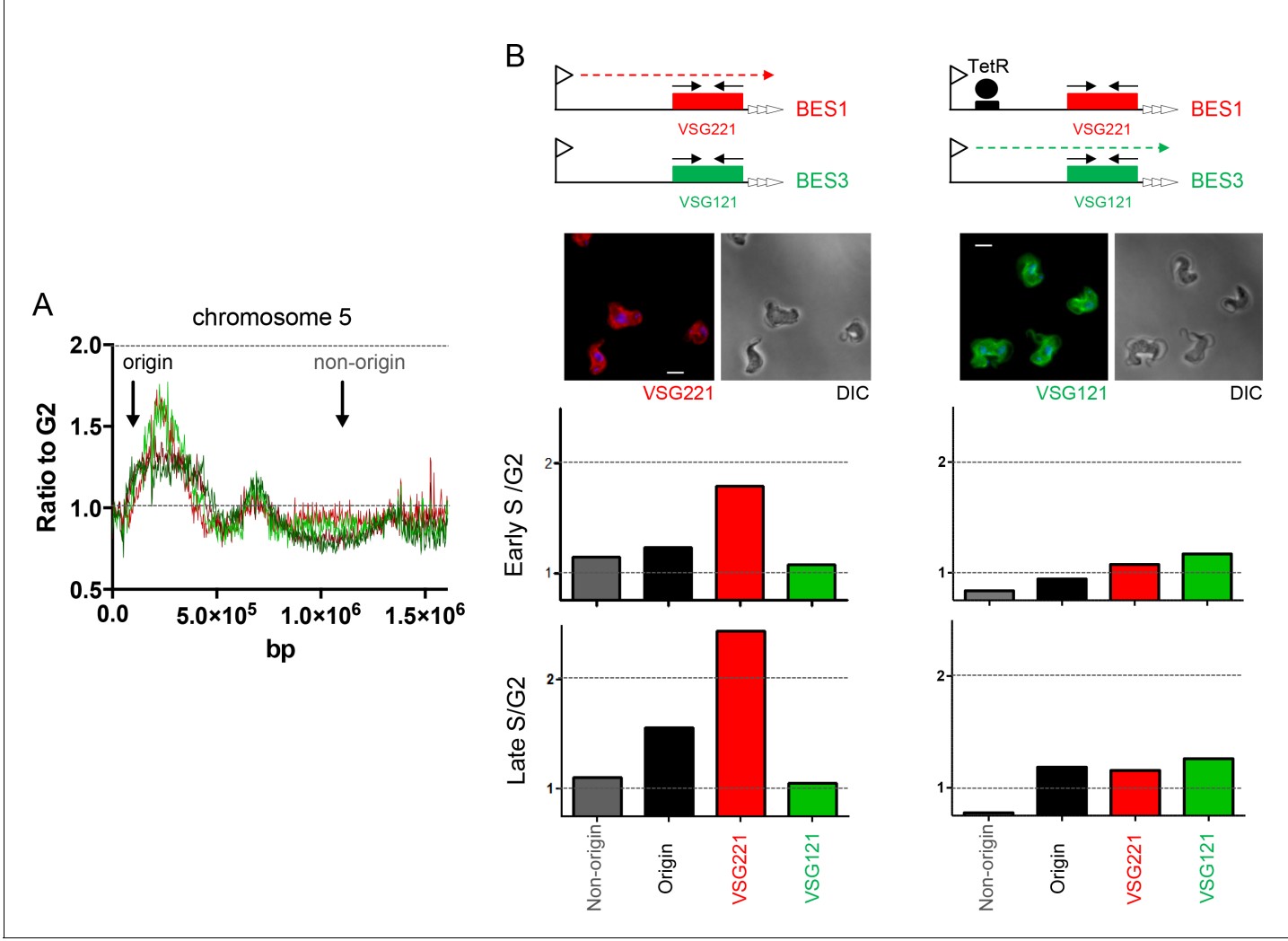

**Figure 8.** Determination of telomere replication timing in *T. brucei* cells expressing distinct bloodstream *VSG* expression sites. (**A**) MFAseq of chromosome 5, as shown in *Figure 6*, comparing S/G2 read depth ratios in 2.5 kbp bins in bloodstream form (early S – light red; late S – dark red) and procyclic form (early S – light green; late S – dark green) cells. Arrows highlight the locations of an early replicating (origin) locus and a late replicating (non-origin) locus, which were used in real-time quantitative (q)PCR validation. (**B**) qPCR to determine replication timing of *VSG221, VSG121* and chromosome 5 origin and non-origin loci in cells in which BES1 (containing VSG221, red box) is actively transcribed (left), or in which (left) elongation of BES1 transcription in blocked by Tet repressor (TetR, black circle) binding to a Tet operator (black box) adjacent to the BES promoter (arrow), leading to transcription (dotted arrow) of BES3 (containing VSG121, green box). In each representation of the BES only the *VSG* genes are shown and black arrows denote the approximate location of primers used in qPCR; below each diagram immunoflouresence microscopy with anti-VSG221 (left, red) and anti-VSG121 antiserum (right, green) is shown (cells are shown by differential interference contrast, DIC). Graphs depict the relative abundance of PCR product from *VSG221, VSG121*, origin and non-origin loci in the two cell types shown above; in each case qPCR was used to determine the amount of the PCR products in DNA from early S phase cells relative to G2 (upper graph), or in late S phase cells relative to G2 (lower graph). S/G2 ratios are the mean of three qPCR repeats.

reaction relative to general HR (which catalyses the change in VSG) and the potential focus on the active BES. VSG switch initiation has been modelled by the direct generation of a DSB in the active BES through the controlled expression and targeting of I-SceI (*Boothroyd et al., 2009*; *Glover et al., 2013*; *Glover and Horn, 2014*), which results in VSG switching. However, to date there is little evidence for the direct formation of a DSB being the initiating strategy in VSG switching, despite data showing that breaks can be detected around the telomeric VSG (though with uncertainty as to whether these are limited to the active BES or to the 70 bp repeats) (*Boothroyd et al., 2009*; *Glover et al., 2013*; *Jehi et al., 2014*). As a result, the nature of the lesions that first form in the BES to drive VSG switching, and the route by which the lesions are generated,

is unclear. Indeed, several initiation models having been proposed (*Barry and McCulloch, 2009*): an uncharacterised endonuclease (*Barry, 1997*), telomere instability (*Dreesen et al., 2007*; *Hovel-Miner et al., 2012*; *Jehi et al., 2014*) and transcription-derived instability (*Kim and Cross, 2010*; *2011*). Here, we have tested the association between I-SceI-mediated DSB formation and VSG switching, revealing that endonuclease mediated DSB formation is unlikely to be the route for initiation of *T. brucei* antigenic variation. Instead, we reveal a remarkably precise association between DNA replication timing and transcription of the single active VSG BES in BSF cells, suggesting that replication may drive VSG switching.

To test the role of endonuclease-generated, BES-focused DSBs in the initiation of VSG switching, we first analysed the genome repair functions of one of two *T. brucei* RecQ-like proteins, TbRECQ2. Several lines of evidence indicate that TbRECQ2 localises to and repairs DNA breaks, including DSBs. First, *T. brucei recq2* null (-/-) mutants display impaired survival in the presence of three compounds that can cause DNA damage, including phleomycin, which is known to generate DSBs (*Falaschi and Kornberg, 1964*; *Reiter et al., 1972*). Second, TbRECQ2 protein relocalises to subnuclear foci after exposure to phleomycin and, critically, in these conditions the putative helicase displays pronounced colocalisation with TbRAD51, an enzyme that binds single-stranded DNA formed at DSBs to catalyse HR repair. Finally, *T. brucei recq2-/-* mutants display reduced survival relative to WT cells after I-SceI induction of DSBs, both in the active BES and elsewhere in the genome, arguing for impaired repair of this lesion. All the above findings are consistent with the activities of related RecQ helicases in other cells, where the enzymes are known to contribute widely to HR. In this regard, the repair and VSG switching (see below) phenotypes observed for TbRECQ2 are highly reminiscent of those described for *T. brucei* Topo3α (*Kim and Cross, 2010*) and RMI1 (*Kim and Cross, 2011*). Thus, it seems plausible that TbRECQ2, and not the other putative *T. brucei* RecQ-like helicase (TbRECQ1), interacts with these factors to form the *T. brucei* homologue of the RTR complex (*Mankouri and Hickson, 2007*).

Given the above evidence for TbRECQ2 repair functions, the enzyme provided a means to test the role of endonuclease-generated DSBs in the initiation of VSG switching. Loss of TbRECQ2 results in an increased rate of VSG switching, as a consequence of altered pathways of recombination: increased levels of telomere exchange and *VSG*-proximal gene conversion. Both these phenotypic outcomes are incompatible with the effects of TbRECQ2 loss on repair of I-SceI-mediated DSBs in the active BES, where the rate of repair is reduced and there is no change in repair pathway. The effects of TbRECQ2 loss on VSG switching are reminiscent of the phenotypes seen after mutation of Topo3α or RMI1 (*Kim and Cross, 2010*;, *2011*), providing more evidence that these factors act together in *T. brucei*. Though VSG switching rates increase in each of the null mutants, the extent of this change is somewhat variable: a 2–3 fold increase in *recq2-/-* mutants, compared with 4-fold and 10–40 fold increases in *rmi1* and *topo3α* null mutants, respectively (*Kim and Cross, 2010*; *2011*). The differing extent of VSG switching increase may be consistent with findings that *S. cerevisiae* Sgs1 and Top3-Rmi1 can act independently on strand exchange intermediates (*Fasching et al., 2015*), but may equally be explicable by subtle differences in the strains used or the growth conditions. In this regard, although the switching rate of the WT cells used by Kim and Cross is broadly similar to the data presented here, the switching profile is somewhat different. For example, in situ switchers were almost entirely absent (<2% of total switchers) in the Kim and Cross studies (*Kim and Cross, 2010*; *2011*), whereas they constituted ~27% of WT switchers here; conversely, ~65% of WT switchers recovered by Kim and Cross used *VSG* GC, as opposed to only ~10% here. The pattern of WT VSG switching described here is very comparable with that seen by *Povelones et al. (2012)*, who used the same constructs but generated the cells independently from us. Thus, it seems likely that small differences in the constructs, their expression levels or their integration into the active BES may result in the WT VSG switching profile differences discussed above. Irrespective of these differences, the increased contribution of VSG GC and telomere XO to VSG switching is a common effect of *recq2-/-* (~50% and ~40%, respectively), *rmi1-/-* (70% and 25%, respectively)and *topo3α-/-* (70% and 23%, respectively) mutation (*Kim and Cross, 2010*; *2011*). Similarly, all three -/- mutants display an almost complete absence of events leading to ES loss (either GC or deletion). As has been argued before (*Kim and Cross, 2010*; *2011*), the increase in telomere XO is striking, as this effect is consistent with the action of the RTR complex in processing recombination intermediates to suppress chromosome crossover (*Cejka et al., 2010*). In yeast, Sgs1 mutation leads to increased crossover recombination after the formation of a chromosome-internal DSB

(*Ira et al., 2003*). The absence of crossovers in HRES *recq2-/-* mutants after I-SceI induction provides further evidence that an induced DSB in the *T. brucei* active BES is tackled by a repair strategy that differs markedly from that which directs VSG switching. Yeast Sgs1 mutants also display increased repair by break-induced replication (BIR) at a telomere-proximal DSB (*Lydeard et al., 2010*). The decreased survival of HRES *recq2-/-* mutants after I-SceI induction indicates that, perhaps surprisingly, an induced DSB in the active BES is inefficiently repaired by BIR, despite the potential for this reaction to direct VSG switching of telomeric genes (*Boothroyd et al., 2009*). Inefficient engagement of a DSB in HR-mediated repair is consistent with the qPCR we describe after I-SceI-mediated cleavage of the BES (see below). In contrast, RTR action on recombination intermediates, including Holliday junctions, that can arise following replication fork stalling (*Mankouri and Hickson, 2007*) may more readily explain the *T. brucei* RTR phenotypes, leading to the suggestion that replication-associated instability initiates VSG switching (below).

Targeted formation of a DSB to elicit recombination and allow temporal analysis of repair has been most extensively described in *S. cerevisiae* (*Hicks et al., 2011*; *Renkawitz et al., 2013*). Such studies suggest that endonuclease cleavage, DNA DSB processing and recruitment of Rad51 occur rapidly, with homology search and functional engagement of a homologous DNA substrate being slower reactions that follow from the break. As has been stated elsewhere (*Glover et al., 2013*), the main effect of an I-SceI-induced DSB in the active BES is cell death, to a larger extent than the same lesion in the interior of chromosome 11. By monitoring the presence of an intact I-SceI site, we show that cleavage occurs in HRES cells at least as rapidly as in HR1, with few cells retaining an intact site after ~8 hr (1 cell cycle). I-SceI cleavage therefore appears to be rapid in *T. brucei* also, with a timing largely consistent with formation of single-stranded DNA, cell cycle impairment and the detection of damage by formation of either RAD51 or γH2A foci (all at around 12 hr) (*Glover et al., 2013*; *Glover and Horn, 2014*). However, whereas loss of PCR-amplifiable I-SceI sequence is maintained in HR1 cells from ~8–72 hr after I-SceI induction, intact I-SceI target sequence reforms at least twice in HRES cells over the 32 hr after induction. After the I-SceI site is maximally cleaved in HR1 cells, recombination product (primarily gene conversion from the unbroken chromosome 11 homologue) gradually forms over the next 72 hr (*Glover et al., 2008*), consistent with induction of repair. In contrast, the abundance of the *VSG* gene (*VSG221*) positioned downstream of the I-SceI site in HRES closely mirrors that of the I-SceI target sequence and, moreover, virtually all HRES cells continue to express VSG221 protein 24 hr after I-SceI induction. Collectively, these data reinforce the evidence for distinct responses to a DSB in the two locations examined (*Glover and Horn, 2014*). In addition, these data reveal that repair efficiency and profile is markedly different in the active BES compared with the interior of chromosome 11. In fact, the data suggest that recovery of VSG switchers following induction of a DSB in the active BES may not result from rapid induction of efficient *VSG*-directed HR. Instead, switchers may be selected: i.e. most HRES cells that suffer a DSB die because repair is inefficient, and the population is gradually replaced thereafter, initially by cells in which the I-SceI site has not been cut and gradually by cells that have undergone a VSG switch that removes the I-SceI target. Given the slow kinetics of repair after Rad51 loading in yeast (*Hicks et al., 2011*), it seems likely that the explanation for the high rate of death after DSB formation in the active *T. brucei* BES is that strand exchange (homology search, invasion or resolution) is very inefficient in this setting. Why this might be awaits further analysis, but it may reflect the limited length or level of sequence identity between *VSG* flanks (including the 70 bp repeats), or the complexity in searching for a repair substrate throughout the *VSG* archive.

What might explain VSG switch initiation, if not the direct formation of a DSB, such as by an endonuclease? MFAseq mapping, validated by VSG-focused qPCR, provides the first evidence that initiation could be linked to DNA replication. Throughout the *T. brucei* genome, we reveal pronounced rigidity in core genome replication timing, with the same origins used with the same efficiency in both BSF and PCF cells. In this context, the actively transcribed BES is unique, being the single mapped telomeric site that is early replicating, and the only locus in the genome that displays different replication timing in PCF and BSF cells. As all BES are silenced and late replicating in PCF cells, the singular early replication of the active BES in mammal-infective *T. brucei* suggests a model for VSG switch initiation in which transcription of the BES allows the site to become accessible for replication and establishes conditions that generate instability, most likely through replication-transcription clashes within the BES (*Figure 9*). Replication stalling can generate the structures that the RTR complex acts upon (*Cejka et al., 2010*; *Mankouri and Hickson, 2007*), explaining the differing

contributions of TbRECQ2 to VSG switching and I-SceI-induced DSB repair. Moreover, stalling of replication by transcription can lead to DNA rearrangements (*Bermejo et al., 2012*) and pausing of the replication fork has been shown to induce targeted mating type switching in fission yeast (*Klar et al., 2014*). How early replication of the active BES in *T. brucei* BSF cells occurs is currently unclear, in particular because replication direction cannot be determined from the MFAseq data. Thus, two possibilities can be considered. In one scenario (*Figure 9B*), replication initiates from the BES promoter environment, which would be compatible with the close association between origin activity and TbORC1/CDC6 binding at transcription start sites in the genome core (*Tiengwe et al., 2012*). Though replication and transcription would be co-directional in this model, clashes between the two processes can occur in this arrangement due to the different rates of the reactions (*Merrikh et al., 2011*). In addition, it is possible that replication or transcription could encounter progression difficulties in traversing the 70 bp repeats, providing some localisation of the clashes, which

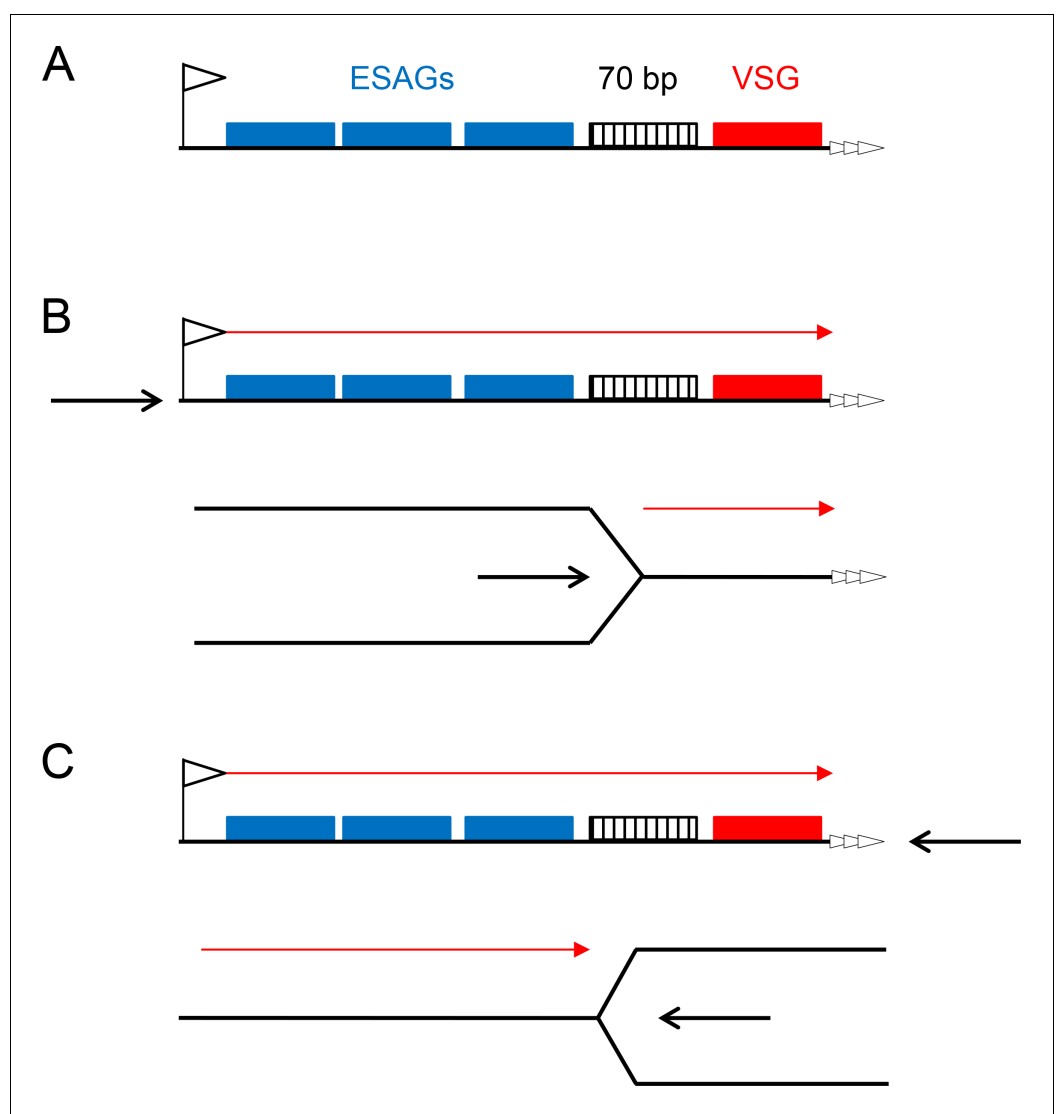

**Figure 9.** Two models for replication-directed VSG switching. A schematic of a bloodstream *VSG* expression site is shown (**A**; not to scale), detailing key features (left to right): the promoter (flag), a number of expression site-asscociated genes (ESAGs; blue boxes), 70 bp repeats (hatched box), the *VSG* gene (red box) and the telomere repeats (white arrows). Transcription direction is detailed in **B** and **C** (red arrow), which compare the effects of replication initating (black arrow) at the promoter or at the telomere, with replication fork movement proceeding left to right, or right to left, respectively.

may lead to DNA breaks, including DSBs. A complication in this model is uncertainty about whether control of BES transcription is exerted by differential promoter activity in the active versus the silent BES (*Kassem et al., 2014*; *Nguyen et al., 2014*). In a second model (*Figure 9C*), replication initiates at the telomere of the BES, leading to head-on collision with transcription; variation in the location of the collisions might explain the uncertainty of DSB mapping in the BES (*Boothroyd et al., 2009*; *Glover et al., 2013*; *Jehi et al., 2014*), and events that lead to gene loss or replacement upstream of the 70 bp repeats. Recently, TbORC1/CDC6 has been documented to bind telomeres in *T. brucei*, but whether this is selective for the active BES, and if it reflects a role in gene silencing or directing replication, is unknown (*Benmerzouga et al., 2013*). In contrast with the data here, previous work has shown that the replicated copies of the active BES are segregated later than silent BES during mitosis (*Landeira et al., 2009*). This observation is not incompatible with unique early replication of the active BES and could, indeed, be explained by a delay in chromatid separation due to the increased presence of unresolved recombination intermediates between sister chromatids at this telomeric locus.

## Materials and methods

### *Trypanosoma brucei* strains, growth and transformation

*T. brucei* BSF cells, strain Lister 427were used throughout and maintained in HMI-9 medium supplemented with 10% (v/v) FBS (Sigma-Aldrich, Missouri, USA) and 1% (v/v) of penicillin-streptomycin solution (Gibco), at 37°C and 5% CO2 in vented flasks. BES1 TetR blockade cells (*Glover et al., 2007*) were generously provided by David Horn and Lucy Glover. Other genetically modified cells were generated by transfection, as described previously (*Burkard et al., 2007*), and clones selected using the following drug concentrations: 10 µg.mLl$^{-1}$ blasticidin, 5 µg.mL$^{-1}$ G418 (Neomycin, NEO), 10 µg.mL$^{-1}$ hygromycin, and 0.2 µg.mL$^{-1}$ puromycin. For VSG switching analysis, the GFP221hygTK cell line was generated based on work and plasmids described in *Povelones et al., (2012)* (constructs 221GP1 and HYG-TK; generous gift, Gloria Rudenko). These cells were cultured in thymidine-free medium: Isocove's Modified Dulbecco's Medium (Gibco) supplemented with 20% FBS (Sigma-Aldrich), 1 mM hypoxanthine, 0.05 mM bathocuproine disulphonic acid, 1 mM sodium pyruvate, 1.5 mM L-cysteine and 200 µM β-mercaptoethanol. Procyclic cell forms (PCF) cells, strain Lister 427, were cultured in SDM-79 (Gibco) supplemented with 10% (v/v) FBS (Sigma-Aldrich), 1% (v/v) penicillin-streptomycin solution (Gibco), and 5 µg.mLl-1 of haemin (Sigma-Aldrich), at 27°C, in non-vented flasks. Cell density was assessed using a Neubauer improved hemocytometer, as standard.

### Genetic manipulation

Heterozygous (+/-) and homozygous (-/-) knockout mutants of TbRECQ2 mutants were generated by deleting most of replacing most of the gene's the open reading frame (ORF; *Figure 1*) with a selective drug marker gene. Two modified versions of the pmtl23 plasmid (gift, Marshall Stark, University of Glasgow), containing either the blasticidin or neomycin resistance genes, were used. In this system, the 5' and 3' flanking non-translated regions of TbRECQ2 ORF were PCR-amplified (5' region – GATCTTCAAGCTTGCGGCCGCTGTGTAAATCCGTTCCTTTCTTC, and GATCTTCTCTAGA TACAACGACACAATACCAACCAC; 3' region – GATCTTCGAGCTCACAGACAATCTCCATCAG-CAACC, and GATCTTCATCGATGCGGCCGCATAAGACATCCACCAGAACCTGC) and cloned in a four-way ligation into the modified pmtl23 plasmid, with each flank surrounding the drug resistance gene. The selective drug marker flanked by the *TbRECQ2* 5' and 3' non-translated regions was then excised using NotI and transfected into BSF cells, and clones selected using 10 µg.ml$^{-1}$ blasticidin or 5 µg.mL$^{-1}$ G418. *Tbrecq2* mutants were analysed by RT-PCR, amplifying a 232 bp region of the *RECQ2* ORF with primers TTTGTGATAACTGCGCAAGC and ACCTTGGAGTGAGCTGAACC; a part of *TbPIF6* was amplified using primers GGTGGGTGTACGATCCATTC and TCGCCAAGGAGAA TAACCTG as a control. RNA was extracted from the cells using the Qiagen RNeasy kit, and cDNA synthesis was performed using random primers and the Primer Design Precision nanoScript Reverse Transcription kit (Primer Design), according to manufacturer's instructions.

In order to N-terminally epitope tag TbRECQ2 with 12myc, a modified version of the construct pEnT6B (*Kelly et al., 2007*) was used. In this case, two fragments were PCR-amplified: a region of the *TbRECQ2* ORF immediately downstream of the start codon, using the primers CAGACTAGTTC

TGTCCACAGAATTCAT (containing an SpeI restriction site) and CAGGGTACCAGGACAAAACAC TAAAAAATA (containing a KpnI site); and a section from the 5′ flanking un-translated region immediately upstream of the *TbRECQ2* ORF, using the primers CAGGGTACCGACAAAGATTTAAG TTGCGTCT (containing a KpnI site) and CAGGGATCCTCGCCGCGGTAATAGTTG (containing a BamHI site). The resulting plasmid was then linearized using KpnI prior to transfection into BSF cells, and transformants were selected with 10 µg.ml$^{-1}$ blasticidin.

For the VSG switching analysis, MITat1.2 BSF cells were first transformed with 221GP1 (*Sheader et al., 2004*) after digestion with NotI and XhoI; transformants were selected with 0.2 µg. mL$^{-1}$ puromycin. These cells were then transformed with the construct HYG-TK, which was digested with NotI and HindIII prior to transformation. Significant difficulty was encountered in propagating the HYG-TK construct without rearrangement, and growth in *E. coli* XL 10 Gold Cells (Stratagene) and ZYM-5 medium appeared to provide greatest stability. Prior to transformation with HYG-TK, the eGFP-PUR cells were cultured in medium lacking thymidine, and transformants were selected in the same medium using 0.2 µg.mL$^{-1}$ puromycin and 10 µg.mL$^{-1}$ hygromycin. Integration of the 221GP1 construct was confirmed by PCR using primers GTGACCACCCTGACCTAC and GCAAACTGTGA TGACCCGC. Integration of the HYG-TK construct was confirmed by PCR using primers TTTACGGGCTACTTGCCATT and CCTCATTTTGGATTTTGCTCCT. Expression of eGFP and VSG221 was confirmed by western blotting (antisera below).

## *T. brucei* RecQ homologue identification and sequence analysis

Standard (default settings) protein-protein Basic Local Alignment Search Tool (BLAST) (blastp) was used to identify potential RecQ helicase-encoding genes in *T. brucei*. Searches were performed using 18 RecQ protein sequences from *Homo sapiens, Mus musculus, Arabidopsis thaliana, Caenorhabditis elegans* and *Saccharomyces cerevisiae* as queries and the *T. brucei* Lister 427 strain genome as target (http://tritrypdb.org/tritrypdb/), revealing two genes. Reciprocal blastp analysis was then performed against the non-redundant protein sequences database (default settings) (http://blast.ncbi.nlm.nih.gov/Blast.cgi) using the predicted protein sequences of both putative *T. brucei* RecQ-like genes; for TbRECQ2 the top hits (lowest E values) were all eukaryotic RecQ helicases. Protein domain analysis of TbRECQ2 predicted sequence was conducted using Pfam, version X (http://pfam.xfam.org/), and InterProScan sequence search (http://www.ebi.ac.uk/interpro/search/sequence-search/).

## Clonal survival after damage

Prior to setting up of the clonal survival assay, all cultures were passaged into drug free medium. The cell cultures were then diluted to 0.5 x 10$^1$ cells.mL$^{-1}$ in media containing either 0.05 µg.mL$^{-1}$, 0.075 µg.mL$^{-1}$ or 0.1 µg.mL$^{-1}$ phleomycin, 0.02 mM, 0.03 mM or 0.04 mM hydroxyurea, or 0.0001%, 0.0002%, 0.0003% or 0.0004% MMS, or none as the untreated control. Each of these cultures were then distributed in 200 µl aliquots into three 96 well plates, and the number of surviving clones quantified after ~10 days growth. The mean survival of the treated samples was determined relative to untreated samples for each damaging agent concentration used and for each cell line, with each of the above experiments repeated at least three times.

## Clonal survival and quantitative PCR after I-SceI induction

Assays were carried out as in (*Glover et al., 2013*). Cells were cultured to mid-log phase (1 x 10$^6$ cells.mL$^{-1}$) in Tet-free medium containing phleomycin, puromycin and hygromycin to maintain the I-SceI genetic components in the cells. For clonal survival, cultures were then diluted to sub-clonal dilutions (HR1 cell lines: 0.15 x 10$^1$ cells.mL$^{-1}$; HRES cell lines: 0.26 x 10$^1$ cells.mL$^{-1}$), divided into two aliquots, and Tet (Calbiochem) added to one (final concentration 2 µg.mL$^{-1}$) to induce I-SceI expression. Cultures were distributed in 200 µL aliquots into 96 well plates (four plates each of uninduced and induced cells). After 7–10 days incubation the number of surviving clones was counted and survival was normalised to uninduced cultures. Mean survival in the induced cells was determined from multiple independent repeats the above experiments. Presence of the I-SceI target and *VSG221* in each cell line prior to and after induction of I-SceI expression was evaluated by quantitative real-time PCR (qPCR). To do this, ~1 x 10$^6$ cells were collected at various time points, and gDNA extracted using the Qiagen Blood and Tissue kit, which was then quantified using the Quant-iT PicoGreen

dsDNA Assay Kit (Life Technologies). Each DNA sample was diluted to 0.2 ng.ul⁻¹ and 1 ng was analysed by qPCR using Precision qPCR MasterMix with SYBR Green and low ROX (Primerdesign), and 6 pmol of primers (Eurofins MWG Operon, Ebersberg, Germany), to a total of 25 µl per reaction. For each pair of primers (below), triplicates of each sample were run per plate (MicroAmp Optical 96-well Reaction Plate, Life Technologies), which were sealed with MicroAmp clear adhesive film (Life Technologies). All experiments were run in a 7500 Real Time PCR system (Applied Biosystems), using the following PCR cycling conditions: 95°C for 10 min, followed by 40 cycles of 95°C for 15 sec and 60°C for 1 min (fluorescence intensity data collected at the end of the last step). Data was then analysed by relative quantification using the ∆∆Ct method (7500 software version 2.3, Applied Biosystems) (*Livak and Schmittgen, 2001*). Abundance of the I-SceI site was evaluated using primer pairs CACAACGAGGACTACACCATC and CGGCCTATTACCCTGTTATCC (HR1 cell line), or GTTG TGAGTGTGTGCTTACC and ATCTAGAGGATCTGGGACCC (HRES). *VSG221* abundance was assessed using primers AGCTAGACGACCAACCGAAGG, and GTTTCCTCTCGCCGTGGTCGC. The generation of HR product in the HR1 cells (data not shown) was determined using primers CACA TTCACTTGACCATTCG and GATGCACTTCGAGAGCGTCAG, which recognise a chromosomal sequence deleted during insertion of the I-SceI target constructs, meaning abundance increases from 1n to 2n after gene conversion from that intact homologue. In all cases, product abundance was determined relative to one of two control loci, which were amplified with primer pairs TC TGAACCCGCGCACTTC and CCACTCACGGACTGCGTTT, or TTGTGACGACGAGAGCAAAC and GAAGTGGTTGAACGCCAAAT.

## Immunofluorescence

Cells were harvested by centrifugation at 405 g for 10 min, and washed with 1x PBS supplemented with 15.7 g/L sucrose and 1.8 g/L glucose, pH 7.4. ~2 x 10⁶ cells were then loaded onto each well of a 12-well glass slide (Menzel-Gläser), pre-treated with Poly-L-lysine (Sigma-Aldrich), and allowed to settle for 5 min. The cells were then fixed with 3.7% paraformaldehyde (PFA) for four minutes, and permeabilised for 10 min with 0.2% Triton X-100 (Promega, in 1x PBS). Next, 100 mM glycine was added and incubated for 5 min, twice. The cells were then washed with 1x PBS twice, 5 min each, and incubated with 1% BSA and 0.2% Tween-20 (Sigma-Aldrich) in 1x PBS, for 1 hr. Afterwards, the cells were incubated for 1 hr with mouse anti-myc antiserum conjugated with Alexa Fluor 488 (Millipore) diluted 1:2000 in 1% BSA and 0.2% Tween-20 in 1x PBS. The cells were washed twice with 1x PBS, after which Fluoromount G with DAPI mounting medium (SouthernBiotech) was added and incubated for 3 min. The slide was then covered with a coverslip and sealed with nail varnish. RAD51 was detected in cells treated in the same way, but using polyclonal anti-RAD51 antiserum as described previously (*Trenaman et al., 2013*). VSG221 was detected using a rabbit α- VSG221 antiserum (gift David Horn, University of Dundee) diluted 1:10000, and Alexa Fluor 594 conjugated goat α- rabbit antiserum (Molecular Probes) diluted 1:1000. EP-procyclin was detected using mouse IgG1 α-EP procyclin antiserum (clone TBRP1/247, Cedarlane) diluted 1:500, and Alexa Fluor 488 conjugated goat α-mouse antiserum (Molecular Probes) diluted 1:1000. Images were acquired and examined as described above. Images were acquired using a Zeiss Axioskop 2 fluorescent microscope attached to an HBO100 lamp and a digital ORCA-ER camara and camera controller (Hamamatsu Photonics), using the Volocity 6.1.1 Cellular and Imaging Analysis software (Perkin Elmer). Images were further analysed using Fiji (http://fiji.sc/Fiji).

## VSG switching analysis

A culture of the *GFP221hygTK* cell growing in thymidine-free medium supplemented with 0.2 µg. mL⁻¹ puromycin and 10 µg.mL⁻¹ hygromycin was passaged to a density of 1 x 10⁴ cells.mL⁻¹ in media lacking hygromycin and incubated for 48 hr to allow VSG switched variants to arise; in some experiments the cells were grown in the absence of puromycin, while in others puromycin (0.2 µg.mL⁻¹) was retained. After 48 hr the cultures were diluted to 2.5 x 10³ cells.mL⁻¹, 5 x 10³ cells.mL⁻¹ or 1.25 x 10⁴ cells.mL⁻¹ in the presence of 4 µg.mL⁻¹ ganciclovir (Sigma-Aldrich) and plated in 200 µl aliquots over three 96 well plates, resulting in 0.5, 1.0 or 2.5 x 10³ cells per well. After 7 days growth multiple surviving clones were randomly selected on a random basis and scaled-up in thymidine- and drug-free medium for further analysis (below). The final number of surviving clones was only assessed after a further 7–10 days. VSG switching frequency was calculated in each experiment by dividing the total

number of surviving clones was divided by number of cells plated, to obtain the number of switching events per cell. This number was then divided by the number of generations in the 48 hr incubation prior to plating the cells, to obtain the VSG switching rate (switchers/cell/generation). The number of generations was calculated for each cell line using the cell density measured at the 48 hr time point. To analyse VSG switching events, the expanded clones were collected and both genomic DNA was extracted, for PCR analysis, and whole cell extracts (2.5 x 10⁶ cells per sample) for western blot analysis. VSG221 was detected using rabbit anti-VSG221 antiserum diluted 1:20.000 (gift, David Horn), while eGFP was detected with a rabbit anti-GFP antiserum (Abcam) diluted 1:5000. Both primary antisera were used in combination with the goat anti-rabbit IgG (H+L) horseradish peroxidase (HRP) conjugate antiserum (Molecular Probes) diluted 1:5000. Ponceau staining of the membrane was carried out to confirm that protein was present on the blot for clones that were VSG221⁻ and eGFP⁻. The presence of *VSG221* and *GFP* genes was assessed by PCR using the primer pairs GCAAGTATATACGCTGAAATAAATCAC and TGTTTGGCTGTTCGCTACTGTGAC (*VSG221*), and C TTCTTCAAGTCCGCCAT and GCTCAGGTAGTGGTTGTC (*GFP*). RNA polymerase I large subunit (Tb427.08.5090) was PCR-amplified as a positive control using the primers CTGGATCCAGCGCCG TTCCACGCGAGA and GACTCGAGCTATCCCCAATCCGTGCCGTCCCG.

## Fluorescence-activated cell sorting (FACS) and MFAseq or qPCR

For each sorting, 3 x 10⁸ cells were collected from an exponentially growing BSF cell culture (~1 x 10⁶ cells.ml⁻¹), and centrifuged for 10 min at 1000 g. Cells were then re-suspended in 25 ml of 1x PBS and centrifuged for 10 min at 1000 g. The pellet was then re-suspended in 500 µl of 1x PBS, and 9.5 ml of 1% formaldehyde (methanol-free, Thermo Scientific, diluted in 1x PBS) was added for 10 min at room temperature. The cells were then centrifuged for 10 min at 1000 g, washed once in 10 ml 1x PBS, and centrifuged again. The pellet was next re-suspended to a concentration of 2.5 x 10⁷ cells.ml⁻¹ in 1x PBS, and was stored protected from light at 4°C overnight. The fixed cells were then centrifuged for 10 min at 1000 g, and incubated in 20 ml of 0.01% Triton X-100 (Promega) in 1x PBS for 30 min at room temperature. Next, the cells were centrifuged for 10 min at 700 g, washed in 20 ml of 1x PBS, and centrifuged again. The resulting pellet was then re-suspended to a concentration of 2.5 x 10⁷ cells.ml⁻¹ in 1x PBS with 10 µg.ml⁻¹ of propidium iodide (PI, Sigma-Aldrich) and 100 µg.ml⁻¹ RNase A (Sigma-Aldrich), and incubated for 1 hr at 37°C, protected from light. For PCF cells, 3 x 10⁸ were collected from an exponentially growing PCF culture (~1 x 10⁷ cells.ml⁻¹) and centrifuged for 10 min at 1620 g. The pellet was then washed in 10 ml of 1x PBS supplemented with 5 mM EDTA (Gibco), and centrifuged for 10 min at 1620 g. Next, the cells were re-suspended in 12 ml of 1x PBS supplemented with 5 mM EDTA, to which 28 ml of 100% ice cold-Methanol was added, in a drop-wise fashion while vortexing gently, so that the final fixing solution was 70% (v/v) Methanol, and the cell concentration was 2.5 x 10⁷ cells.ml⁻¹. The cells were then kept at 4°C, protected from light, from overnight up to three weeks. For each FACS sorting session, four FACS tubes (Becton Dickinson) were prepared, each starting with ~1 x 10⁸ fixed cells. The cells were collected and centrifuged for 10 min at 1000 g, at 4°C, washed in 1 ml of 1x PBS supplemented with 5 mM of EDTA, and centrifuged again for 10 min at 1000 g, at 4°C. The pellet was then re-suspended in 4 ml of 1x PBS supplemented with 5 mM EDTA, 10 µg.ml⁻¹ PI and 10 µg.ml⁻¹ RNase A, and incubated for 45 min at 37°C, in the dark. The cells (either BSF or PCF) were then transferred to a FACS tube through a cell strainer cap (BD Biosciences), and sorted into G1, early S, late S and G2 phases using a BD FACSAria I Cell Sorter (BD Biosciences). The sorted cells were collected at 4°C into new FACS tubes containing 200 µl of lysis buffer (1 M NaCl, 10 mM EDTA, 50 mM Tris-HCL pH 8.0, 0.5% SDS, 0.4 mg.ml-1 Proteinase K, and 0.8 µg.ml-1 of Glycogen). After the sorting has been completed, the collected cells were then incubated for 2 hr at 55°C, and the lysate was stored at -20°C. Genomic DNA was extracted using a Blood and Tissue DNA extraction kit (Qiagen), by omitting the lysis steps of the manufacturer's protocol. Sequencing was performed by Eurofins Genomics (Germany); the DNA library was prepared using the TruSeq DNA Sample Preparation kit (Illumina), and sequenced using Illumina HiSeq paired- end 100 bp sequencing system (Illumina). The samples were multiplexed, with each of the early S, late S, and G2 phase samples library DNA, both from BSF and PCF, being processed in the same run, for ease of comparison.

Data from the sequencing was first analysed for quality using FastQC (http://www.bioinformatics. babraham.ac.uk/projects/fastqc/), and then trimmed using fastq-mcf (http://code.google.com/p/ea-utils), to exclude the adapter sequences used during the library preparation and sequencing. The

reads were then aligned to the reference genome (*T. brucei* Lister 427, retrieved from TriTrypDB version 8.0), using Bowtie2 (version 2.2.0 –very-sensitive-local -k1)(*Langmead and Salzberg, 2012*). The aligned reads were then compared using a method adapted from the one described previously (*Tiengwe et al., 2012*), but simplified to facilitate inter-species comparisons. Briefly, the reads were binned in 2.5 Kbp sections along each chromosome, and the number of reads in each bin was then used to calculate the ratios between early S and G2, as well as between late S and G2 samples, scaled for the total size of the read library (reads per 2.5 Kbp per million reads mapped). These data were then represented in a graphical form using Prism 6 (GraphPad software Inc.). Shell scripts used to generate these data are freely available in BitBucket (https://bitbucket.org/WTCMPCPG/tb_antigenic_variation). Sequences are available at the European Nucleotide Archive: PRJEB11437.

MFA by qPCR was performed as described in (*Marques et al., 2015*) with primers targeting *VSG221* (AGCAGCCAAGAGGTAACAGC and CAACTGCAGCTTGCAAGGAA), *VSG121* (AGGAAGGCAAATACGACCAG and TTTGCGGGTAAAAGTCCTTG) and selected origin (TCCCAGAAACCAACTTCAGC and AGTTGGATTGCCATGTCCTC) and non-origin regions (GGCTGGATGATGAGAGGAAC and CCTCCAACCTCAAGATACGC) in chromosome 5. For normalization, a non-origin region in chromosome 2 was used (CTCGCTCTCCGTACAGTTG and CACTCGTCGATGCAACCTC). For each sample, 0.18 ng of gDNA was used, and the data shown are averages of itriplicate experiments.

## Acknowledgements

This work was supported by the Wellcome Trust [089172, 093589], the BBSRC [BB/K006495/1] and Fundação para a Ciência e Tecnologia (FCT, Portugal) [SFRH/BD/68784/2010]. The Wellcome Trust Centre for Molecular Parasitology is supported by core funding from the Wellcome Trust [104111]. We thank Gloria Rudenko, Anna Trenaman and Megan Povelones for the gift of plasmids and help with setting up the switching assay, Dave Horn and Lucy Glover for providing and describing the TetR blockage cells, Diane Vaughan for help with FACS, Lucy Glover and Fernando Fernandez-Cortes for the VSG221 and EP-procyclin immunofluorescence protocols, and many colleagues for discussions, in particular David Horn, Lucy Glover, Sam Alsford, Sebastian Hutchinson and Carlos Machado, as well as all McCulloch and Mottram lab members.

## Additional information

### Funding

| Funder | Grant reference number | Author |
|---|---|---|
| Wellcome Trust | PhD studentship, 093589 | Rebecca Devlin |
| Fundação para a Ciência e a Tecnologia | SFRH/BD/68784/2010 | Catarina A Marques |
| Wellcome Trust | Project grant, 089172 | Richard McCulloch |
| Biotechnology and Biological Sciences Research Council | Project grant, BB/K006495/1 | Richard McCulloch |
| Wellcome Trust | Centre award, 104111 | Richard McCulloch |

The funders had no role in study design, data collection and interpretation, or the decision to submit the work for publication.

### Author contributions

RD, CAM, DP, Conceived and designed the experiments, Conducted the experiments, Analysed the data, Wrote the manuscript; MP, ACZ-L, SJC, Conceived and designed the experiments, Conducted the experiments, Analysed the data; CL, Conducted the experiments, Analysis and interpretation of data; ND, Conceived and designed the experiments, Analysed the data, Drafting or revising the article; RM, Conceived and designed the experiments, Analysed the data, Wrote the manuscript

Author ORCIDs

Catarina A Marques, http://orcid.org/0000-0003-1324-5448

Richard McCulloch, http://orcid.org/0000-0001-5739-976X

## Additional files

### Major datasets

The following dataset was generated:

| Author(s) | Year | Dataset title | Dataset URL | Database, license, and accessibility information |
|---|---|---|---|---|
| Devlin R, Marques C, Prorocic M, Lapsley C, Dickens N, McCulloch R | 2015 | Mapping replication dynamics in Trypanosoma brucei reveals a link with telomere transcription and antigenic variation | http://www.ebi.ac.uk/ena/data/search?query=PRJEB11437 | Publicly avilable at European Nucleotide Archive (accession no: PRJEB11437) |

The following previously published dataset was used:

| Author(s) | Year | Dataset title | Dataset URL | Database, license, and accessibility information |
|---|---|---|---|---|
| Tiengwe C, Marcello L, Farr H, Dickens N, Kelly S, Swiderski M, Vaughan D, Gull K, Barry JD, Bell SD, McCulloch | 2012 | Procyclic Form Origins of Replication - early-mid S phase vs G2 | http://tritrypdb.org/cgi-bin/gbrowse/tritrypdb/?start=1;stop=1193948;ref=Tb927_02_v5.1;width=800;version=100;flip=0;grid=1;id=a1-f69eddbfea76dc039d-b8909290a05d;l=tbru-TREU927_originsOfRepli-cation_McCulloch_Ori-gins_RSRCG2S%1EGene | Publicly available at TriTrypDB |

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
