## [Decision Letter]

Thank you for submitting your work entitled "Mapping replication dynamics in *Trypanosoma brucei* reveals a link with telomere transcription and antigenic variation" for consideration by *eLife*. Your article has been reviewed by three peer reviewers, and the evaluation has been overseen by a Reviewing Editor and Richard Losick as the Senior Editor.

The reviewers have discussed the reviews with one another and the Reviewing Editor has drafted this decision to help you prepare a revised submission.

General assessment:

The manuscript under consideration from Devlin and colleagues addresses the fascinating problem of antigenic variation in *T. brucei* ("VSG Switching"), which is at the heart of this parasites ability to cause disease in humans and other mammals. The manuscript attempts to clarify the important issue of how the mechanism of coat variation is initiated. The authors report the very interesting result that while origins of replication are conserved (presumably in location and firing) in the two trypanosome life forms (PC and BF), the lone exception is the active BES in the BF which replicates early. From there, the implication is that replication is key for the initiation of VSG switching, and the authors go some way toward showing that. It's not an unexpected result, necessarily (it was one of the mechanisms contemplated by many in the field) but it is really nice to see it substantiated experimentally. This could well be the mechanism that produces the DNA DSB that is an obligate intermediate of any cut and paste recombination event.

And really, that's the only (but significant) problem with the paper: that they try to "invalidate" a substantial body of data showing that this is the case, through experiments that are either over interpreted, or potentially misinterpreted. Because indeed, it is rather likely that the recQ mutant accumulates DNA damage, some of which could be a direct DSB or a SSB that develops into a DSB. The authors might want to examine DSB formation in the mutants, for instance to demonstrate this.

The recommendation therefore will be for "major revision", ideally with the main focus being on the robust experimental data implicating replication (and RECQ2, and transcription fork stalling, potentially) as an initiating event to VSG recombination. This revision should be chiefly textual but an also include additional experiments, as summarized below.

Summary of main concerns:

1) The loss of RECQ2 resulted in fewer survivors in HR1 and HRES cells, but RECQ2 null cells have higher VSG switching rate than WT cells. The authors therefore concluded that DSBs are not linked with the initiation of VSG switching. In my opinion, the observation that RECQ2 null cells have higher VSG switching rate is consistent with the idea that DSBs induce VSG switching. It is important to note that RECQ2 may also play an important role in DNA replication in addition to DNA damage repair. Therefore, loss of RECQ2 is expected to result in more DSBs (due to less efficient repair of DSBs or due to increased DSB formation resulting from more replication fork stalling or due to both reasons), which in turn can lead to more frequent VSG switching. The key experiment here is to examine whether RECQ2 null cells have more DSBs at subtelomeres or not. This can be done by LMPCR or IF/ChIP using γH2A/RAD51 antibodies. In addition, it is important to test whether RECQ2 plays an important role in DNA replication. If RECQ2 null cells have more replication fork stalling induced DSBs, switchers can arise from recombination events different than those observed in HRES cells, where repair of an induced DSB is assayed.

2) The second concern is about the I-SceI induction in HRES cells. This essentially repeated the same experiment done in Glover et al. 2013, however, with a different outcome. In Glover et al. 2013, all survivors from cells with an I-SceI cut between 70 bp repeats and the active VSG gene are switchers, while in the current study many survivors did not switch. The authors speculated that VSG2 expressers were simply not digested by I-SceI. Therefore, the I-SceI digestion seemed not as efficient as it was in Glover et al. 2013, which was more than 50% (by γH2A IF analysis). However, it is not shown how efficient the I-SceI digestion is induced in this manuscript. It is important to determine the I-SceI digestion efficiency as a control and equally important to increase the induction efficiency to ensure a proper assay. On the other hand, it is possible that the I-SceI cut was efficiently repaired if the digestion occurs in G2 phase where a sister chromatid is available. It would be better if the authors can specifically induce I-SceI cut in G1 and G2 phases separately, but this possibility should be at least discussed. In addition, possible reasons for the different results of this study and the one in Glover et al. 2013 should be discussed.

3) DNA fragility resulting from collisions between DNA replication and transcription machinery is an exciting hypothesis (supported by data in other organisms) that has been asserted previously in *T. brucei* switching and the data presented in Figure 7 in support of this prediction a potential strength of this manuscript that should have been explored further. However, DNA fragility in the actively expressed subtelomere doesn't directly argue against "direct formation of DSBs.… [as] the initiating event in VSG switching," as the authors would suggest. Based on the observations, it seems equally likely that DNA replication through the actively transcribed region is a challenge that requires RECQ2 function, while loss of RECQ2 likely causes more replication fork stalling, resulting in more DSBs and leading to more VSG switching subsequently. It is therefore important to examine DNA replication in the active ES in RECQ2 null cells.

[Editors' note: further revisions were requested prior to acceptance, as described below.]

Thank you for resubmitting your work entitled "Mapping replication dynamics in *Trypanosoma brucei* reveals a link with telomere transcription and antigenic variation" for further consideration at *eLife*. Your revised article has been favorably evaluated by Richard Losick (Senior editor), a Reviewing editor, and three reviewers.

The manuscript has been much improved but there are some remaining issues (noted by reviewers 1 & 2) that need to be addressed with textual changes before formal acceptance, as outlined below (reviewer 3 had no additional concerns).

Reviewer #1:

While the newly added data are compelling, the revision under consideration does not address a major concern that the data are presented in a somewhat confused manner. For instance, the wording regarding how breaks arise and whether they are made "directly" is my biggest concern. After reading both drafts of the manuscript multiple times and the response to reviewers, I now understand that by "directly" the authors are speaking of a hypothesis that a specific endonuclease could initiate DNA break formation and switching. This is problematic for two reasons:

1) There is no longer a "prevailing hypothesis" that a specific endonuclease has a role in switch initiation. There are numerous studies and reviews that either show switching arising from DNA fragility (non-enzymatic) or suggest that DNA replication, transcription, or other forms of site instability are the basis for the DNA lesions that resulting VSG switching. Nonetheless, it is completely reasonable to have this be the main question the paper is addressing, as long as it is clear.

2) The language around this whole subject in the manuscript is confounding and open to misinterpretation. In the author's response, they clarify this issue in the following statement, "The implications of this analysis are important. For instance, it would erroneous for researchers to hunt for a *T. brucei*-encoded endonuclease that cleaves the BES; in addition, data from the use of ISceI-mediated cleavage to model VSG switch initiation must be interpreted with caution, as it may not recapitulate the 'natural' route of initiation".

This is a very important point and should be stated in the manuscript. Also, in the manuscript Introduction (third paragraph) the authors clearly describe the problem they are seeking to address and in essence define what is meant by "directly." But the word "directly" is not used in that passage to clarify their word choice. If directly were defined here many of the difficulties with this paper might be cleared up. Are breaks formed by recombination or transcriptional instability occurring "indirectly", only if they clearly defined as such.

This is not a trivial issue as the use of vague language in the Abstract could lead to a complete misunderstanding by some readers who would read it to think the authors propose breaks are not involved at all.

Recommendation #1:

Use of the word "directly" should be removed from the Abstract and replaced with clearer language. Then it can be defined in the Introduction and used thereafter.

Second major concern:

It is confounding that a paper built on high-quality data and presenting interesting findings is constructed in a manner that distracts from the findings. Specifically, (from the first paragraph of subsection “A DNA doublestrand break in the VSG expression site is inefficiently repaired”), there is nothing "dichotomous" about the observation that RECQ2 knock-down increases sensitivity to DNA damage and increases switching. If fact, the entire body of data in this paper consistently suggest that RECQ2 has a role in DNA damage repair, which may be linked to specific aspects of DNA replication.

Recommendation #2:

The question the paper aims to address should be more clearly stated. If the real question is "how VSG switching is initiated" or "what is the initiating lesion that activates recombinatorial switching" then the authors should state this directly. If they think a major conclusion from these findings is that it is unlikely that a site specific endonuclease is involved they should discuss this in the Introduction and possibly mention it clearly in the Abstract.

The data in each Results section should be presented without interpretation. The conclusions and interpretations reserved for the Discussion. This approach would strengthen the manuscript by highlighting the quality of the data and deemphasizing the existing over-interpretations.

While the data and discoveries are (and previously were) publication worthy, in my view, the manuscript needs a major rewrite with a change of tone and emphasis on the data that is presented and the specific question the authors aimed to address.

*Reviewer #2:*

This revised manuscript has improved greatly. The authors have added several key new results and improved the discussions.

Specifically regarding the authors' conclusion that "phenotypes from RECQ2 null cells argue against models for VSG switch initiation based upon direct DNA break formation", based on the authors' rebuttal and more explanation, this reviewer feels what the authors want to state is that DNA breaks can form due to replication stress, which in turn can induce VSG switching. Therefore, direct DNA break formation, such as a consequence of a specific endonuclease activity, is unlikely the first step of VSG switching. In this regard, this reviewer agrees with the authors. However, this reviewer feels that the term "direct DNA break formation" may not have explained this clearly enough. At this stage, just a few more sentences to clearly describe what do the authors mean about "direct DNA break formation" should suffice.

This reviewer appreciates the authors now have performed additional experiments to confirm that when VSG221 is active, it is replicated early, but when it's silenced, it is replicated late. This is a very strong piece of evidence supporting the hypothesis that replication timing is linked with transcription state. The MFA data on VSG121, however, is not as strong. The fact that there are several genomic copies of VSG121 and the fact that VSG121 may not be the sole active VSG among the analyzed population make it hard to interpret the data. It is probably better not to extrapolate too much.

---

## [Author Response]

*General assessment:*

The manuscript under consideration from Devlin and colleagues addresses the fascinating problem of antigenic variation in T. brucei ("VSG Switching"), which is at the heart of this parasites ability to cause disease in humans and other mammals. The manuscript attempts to clarify the important issue of how the mechanism of coat variation is initiated. The authors report the very interesting result that while origins of replication are conserved (presumably in location and firing) in the two trypanosome life forms (PC and BF), the lone exception is the active BES in the BF which replicates early. From there, the implication is that replication is key for the initiation of VSG switching, and the authors go some way toward showing that. It's not an unexpected result, necessarily (it was one of the mechanisms contemplated by many in the field) but it is really nice to see it substantiated experimentally. This could well be the mechanism that produces the DNA DSB that is an obligate intermediate of any cut and paste recombination event.

We thank the referees for their enthusiasm regarding the striking observation that the actively transcribed BES is the sole early-replicating telomere, and that this effect is limited to BSF cells and not seen in PCF cells (where all BES are silenced). Though the referees are correct that a link with replication has been noted in the field, in fact such contemplation has had little foundation, since we had essentially no knowledge of the replication dynamics of *T. brucei* chromosome (sub)telomeres. Our new data fills this gap in knowledge, but it is important to note (as we have tried hard to stress in the paper) that the potential link between BES replication dynamics and VSG switching remains an association (see below). We have gone to some lengths to emphasise there is considerable uncertainty in the nature of the BES replication (please see the model that we propose, and discussion below), and that extensive and challenging experimental tests of this new hypothesis are needed in the future.

And really, that's the only (but significant) problem with the paper: that they try to "invalidate" a substantial body of data showing that this is the case, through experiments that are either over interpreted, or potentially misinterpreted. Because indeed, it is rather likely that the recQ mutant accumulates DNA damage, some of which could be a direct DSB or a SSB that develops into a DSB. The authors might want to examine DSB formation in the mutants, for instance to demonstrate this.

We’re a little bemused by the suggestion that we are attempting, in the analysis of RECQ2, to ‘invalidate’ previous data. We have no wish to invalidate peer-reviewed, published work. As we have stated, the published data linking DSBs with initiation of antigenic variation are association studies: detection of DNA damage by ligation-mediated PCR (LMPCR) in the BESs, and the demonstration that endonuclease-mediated DSBs can lead to VSG switching. Association studies can only allow the formulation of a hypothesis that DSBs are linked to VSG switch initiation. Thus, the state of the field is that the prevailing hypothesis needs to be critically tested (just as we have stressed above for the link between the new replication data and VSG switching). As an example of why experimental testing needs done, the available data cannot determine the route(s) by which DSBs arise in the BES. In analysing the role of RECQ2, we provide the first test of whether directly formed DSBs are a likely initiating lesion. Our conclusions are not, as the referees suggest (see below, comment 1), that ‘DSBs are not linked with the initiation of VSG switching’. Instead, we have stated that the inconsistency between the effects of RECQ2 mutation on repair of induced DSBs and on the rate and profile of VSG switching indicates that if DSBs are the initiating lesion (which remains entirely possible), it is unlikely they arise directly. The implications of this analysis are important. For instance, it would erroneous for researchers to hunt for a *T. brucei*-encoded endonuclease that cleaves the BES; in addition, data from the use of ISceI-mediated cleavage to model VSG switch initiation must be interpreted with caution, as it may not recapitulate the ‘natural’ route of initiation (though it remains a valuable strategy to model VSG switch dynamics).

Given our RECQ2 data suggest DSBs do not to arise directly it became important we provide an alternative model for switch initiation might occur – hence why we explored replication dynamics of the BES. Again, to be absolutely clear, we are well aware the available data linking BES replication and VSG switch initiation is also an association, and we have tried hard to stress this new hypothesis needs critical testing.

The recommendation therefore will be for "major revision", ideally with the main focus being on the robust experimental data implicating replication (and RECQ2, and transcription fork stalling, potentially) as an initiating event to VSG recombination. This revision should be chiefly textual but an also include additional experiments, as summarized below.

Summary of main concerns:

1) The loss of RECQ2 resulted in fewer survivors in HR1 and HRES cells, but RECQ2 null cells have higher VSG switching rate than WT cells. The authors therefore concluded that DSBs are not linked with the initiation of VSG switching. In my opinion, the observation that RECQ2 null cells have higher VSG switching rate is consistent with the idea that DSBs induce VSG switching. It is important to note that RECQ2 may also play an important role in DNA replication in addition to DNA damage repair. Therefore, loss of RECQ2 is expected to result in more DSBs (due to less efficient repair of DSBs or due to increased DSB formation resulting from more replication fork stalling or due to both reasons), which in turn can lead to more frequent VSG switching. The key experiment here is to examine whether RECQ2 null cells have more DSBs at subtelomeres or not. This can be done by LMPCR or IF/ChIP using γH2A/RAD51 antibodies. In addition, it is important to test whether RECQ2 plays an important role in DNA replication. If RECQ2 null cells have more replication fork stalling induced DSBs, switchers can arise from recombination events different than those observed in HRES cells, where repair of an induced DSB is assayed.

As stated above, nowhere do we conclude that ‘DSBs are not linked with the initiation of VSG switching’. Instead, our conclusion is that ‘the direct formation of a DSB is unlikely to be the route for initiation of *T. brucei* antigenic variation’. In the final Discussion paragraph, we attempted to discuss how replication stalling may link to previous LMPCR data that detected DNA breaks within the BESs (Boothroyd et al., 2009; Glover et al., 2013; Jehi et al., 2014), but we acknowledge we failed to state that stalling may result in DSBs. As a result, we have modified the text:

“In addition, it is possible that replication or transcription could encounter progression difficulties in traversing the 70 bp repeats, providing some localisation of the clashes, which may lead to DNA breaks, including DSBs.”

“In a second model (Figure 8), replication initiates at the telomere of the BES, leading to head-on collision with transcription; variation in the location of the collisions might explain the uncertainty of DSB mapping in the BES (Boothroyd et al., 2009; Glover et al., 2013; Jehi et al., 2014).”

At this stage, it is a major undertaking to evaluate DSB quantity within the BES. First, these loci are highly homologous, seriously compromising any attempts at chIP mapping. Second, our attempts to date at chIP with anti-RAD51 antiserum suggest it is not chIP-compatible (data not shown). Third, we are unconvinced that LMPCR is capable of providing quantification of lesion abundance, given it is based upon end-point PCR. Thus, to ask if RECQ2 mutants, in general, display altered replication or contain more lesions than WT, we provide two new sets of data.

First, we compared the cell cycle distribution of WT cells and the RECQ2 mutants by DAΠ staining, and could find no compelling evidence the mutants displayed an alteration relative to WT, suggesting that lack of RECQ2 does not result in detectable stalling at a discernible cell cycle stage. These data are added as a new panel to Figure 1. We have added the following text to the paper to refer to these data:

“To ask if the growth change [in the mutants] results from an impediment in completing a cell cycle stage or traversing between sequential stages, cells were stained with DAΠ to visualise nuclear (N) and kinetoplast (K) DNA. […] Thus, the growth impairment of the RECQ2 mutants is not due to detectable stalling at a discernible cell cycle stage or transition.”

Second, we have used western blotting to evaluate levels of γ H2A (thr130 phosphorylated H2A) and find evidence there is increased signal intensity of the repair-associated histone modification in the *recq2*-/- mutants, indicating greater levels of endogenous nuclear damage. These data are added as a new panel to Figure 1 and we have added the following text:

“Western blotting to detect the levels of Thr130 phosphorylated histone H2A (γ-H2A) revealed an increased signal in the *recq2-/-* mutants relative to WT (Figure 1), indicating this modification accumulates in the absence of the helicase to an extent comparable to that seen in WT or -/- mutants cells grown for 18 hr in 0.0003% MMS (see below). […] Thus, accumulation of the histone variant in the *recq2-/-* mutants indicates an increased level of nuclear DNA damage, which appears not to impair cell cycle progression but may impede cell growth or survival.”

2) The second concern is about the I-SceI induction in HRES cells. This essentially repeated the same experiment done in Glover et al. 2013, however, with a different outcome. In Glover et al. 2013, all survivors from cells with an I-SceI cut between 70 bp repeats and the active VSG gene are switchers, while in the current study many survivors did not switch. The authors speculated that VSG2 expressers were simply not digested by I-SceI. Therefore, the I-SceI digestion seemed not as efficient as it was in Glover et al. 2013, which was more than 50% (by γH2A IF analysis). However, it is not shown how efficient the I-SceI digestion is induced in this manuscript. It is important to determine the I-SceI digestion efficiency as a control and equally important to increase the induction efficiency to ensure a proper assay. On the other hand, it is possible that the I-SceI cut was efficiently repaired if the digestion occurs in G2 phase where a sister chromatid is available. It would be better if the authors can specifically induce I-SceI cut in G1 and G2 phases separately, but this possibility should be at least discussed. In addition, possible reasons for the different results of this study and the one in Glover et al. 2013 should be discussed.

I’m afraid that we don’t agree with the above interpretation of our findings, which in fact are very comparable with the data for HRES cells described by Glover and colleagues in both their 2013 (PLoS Pathogen) and 2014 (Nucleic Acid Res) papers.

A) *In Glover et al. 2013, all survivors from cells with an I-SceI cut between 70 bp repeats and the active VSG gene are switchers, while in the current study many survivors did not switch.* This statement is incorrect. In our study, as in Glover et al. 2013, all analysed clonal survivors after ISceI induction of WT HRES cells had switched the expressed VSG. In Figure 4 we perform PCR on 25 HRES WT survivors and show that none retained the VSG221 gene, meaning they must be expressing another VSG; Glover et al. found, by the same PCR, that 1/22 HRES survivors retained the VSG221 gene, though no longer expressed the protein. An anomaly in our study is that 4% of the HRES WT survivors were still resistant to puromycin, for reasons that are unclear.

B) *The authors speculated that VSG2 expressers were simply not digested by I-SceI.* This statement appears to be referring to results for the RECQ2 mutants, where we were attempting to explain the greater numbers of puromycin-resistant HRES RECQ2 mutant clones after ISceI induction. On re- reading, we can see that this was unclear and have therefore changed the wording:

“These data are most simply explained by a greater number of HRES *recq2-/-* mutants being recovered (relative to HR1 mutants) in which a DSB has not been induced, reflecting the very limited survival capacity of *recq2-/-* mutants after a DSB is made in the active BES.”

C) *Therefore, the I-SceI digestion seemed not as efficient as it was in Glover et al. 2013, which was more than 50% (by γH2A IF analysis). However, it is not shown how efficient the I-SceI digestion is induced in this manuscript. It is important to determine the I-SceI digestion efficiency as a control.* In fact, we directly assess the efficiency of ISceI cleavage using a qPCR assay (Figure 5). These data reveal 80-100% of cells do not have intact (PCR-amplifiable) ISceI sequence 2-8 hrs after Tet addition, arguing that cleavage is extremely efficient (in fact, more efficient than in HR1 cells). This seems to us a more direct measure of DNA cleavage than assessing levels of modified histone, though we find (Devlin and McCulloch, unpublished) that 12 and 24 hrs after Tet addition the average signal intensity of γH2A (evaluated by IF) in HRES cells increases around 4-fold relative to uninduced HRES cells. This finding appears very comparable with the increase in γH2A foci from 10% in uninduced HRES cells to around 50% in HRES cells 12 hr after Tet addition (Glover et al., 2014).

In summary, we suggest that our data display good overlap with those described previously by Glover et al., both in terms of efficiency of ISceI-mediated BES cleavage and induction of switching by loss of VSG221. Just as importantly, we also observe the main response to a DSB mediated by ISceI in the active BES is cell death, to a very similar extent to that described by Glover et al.

3) DNA fragility resulting from collisions between DNA replication and transcription machinery is an exciting hypothesis (supported by data in other organisms) that has been asserted previously in T. brucei switching and the data presented in Figure 7 in support of this prediction a potential strength of this manuscript that should have been explored further. However, DNA fragility in the actively expressed subtelomere doesn't directly argue against "direct formation of DSBs.… [as] the initiating event in VSG switching," as the authors would suggest. Based on the observations, it seems equally likely that DNA replication through the actively transcribed region is a challenge that requires RECQ2 function, while loss of RECQ2 likely causes more replication fork stalling, resulting in more DSBs and leading to more VSG switching subsequently. It is therefore important to examine DNA replication in the active ES in RECQ2 null cells.

In the above quote, the statement we make is taken out of context. The full statement is: “We show that loss of TbRECQ2 impairs DSB repair, consistent with the observation that the protein localizes to such lesions. Conversely, TbRECQ2 mutants display elevated rates of VSG switching, indicating it is unlikely that the direct formation of DSBs is the initiating event in VSG switching.” We are specifically referring to the RECQ2 data, which argue against VSG switch initiation through direct formation of a DSB, such as occurs by ISceI cleavage.

It is important to be clear about what we can, and what we cannot say from the available data. Our data argue that RECQ2 acts in VSG switching, consistent with the widespread role for homologous recombination in this immune evasion process. However, our data also show that the mode of action of RECQ2 in VSG switching is inconsistent with how the *T. brucei* protein (indeed all RecQ homologues) acts upon a directly generated DSB. Thus, we should discount (or at the very least question) the direct formation of a DSB as being the route to VSG switch initiation. It is of course possible that ‘loss of RECQ2 likely causes more replication fork stalling, resulting in more DSBs and leading to more VSG switching subsequently’. However, we do not yet know the direction, rate or initiation site of replication in the BES; nor do we know if replication stalls somewhere in the BES, or what structures form if stalling occurs. In our view, it is overly speculative to make any such assertions at this stage, and hence we have proposed a limited model (Figure 9) in which we have been circumspect in how we discuss the mode of action of RECQ2 in VSG switching and the proposed link between replication and VSG switching.

We agree that *it is therefore important to examine DNA replication in the active ES in RECQ2 null cells*: examining replication dynamics, both in WT and RECQ2 mutant cells, will provide critical tests of the hypothesis we propose, just as examining RECQ2 has provided a test of the hypothesis that directly-formed DSBs initiate VSG switching. However, I’m sure the referees appreciate that such tests are a long-term endeavour, necessitating new single molecule approaches to evaluate BES replication dynamics, since MFAseq is too crude for such analyses.

[Editors' note: further revisions were requested prior to acceptance, as described below.]

The manuscript has been much improved but there are some remaining issues (noted by reviewers 1 & 2) that need to be addressed with textual changes before formal acceptance, as outlined below (reviewer 3 had no additional concerns).

Reviewer #1:

While the newly added data are compelling, the revision under consideration does not address a major concern that the data are presented in a somewhat confused manner.

[…]

*This is not a trivial issue as the use of vague language in the Abstract could lead to a complete misunderstanding by some readers who would read it to think the authors propose breaks are not involved at all.*

*Recommendation #1:*

Use of the word "directly" should be removed from the Abstract and replaced with clearer language. Then it can be defined in the Introduction and used thereafter.

A) We have restructured the final paragraph of the Introduction to provide greater clarity:

“Despite the emerging association between DNA DSBs and VSG switching, questions remain about the detailed mechanism(s) of VSG switch initiation. For instance, are DSBs generated directly in the active BES, such as through the action of an endonuclease, as occurs during S. cerevisiae mating type switching (Lee and Haber, 2015)?[…] Second, we provide evidence for strong association between replication timing and BES transcription, indicating that VSG switch initiation may be mechanistically linked to DNA replication.”

B) We have altered the wording of the Abstract:

“Survival of *Trypanosoma brucei* depends upon switches in its protective Variant Surface Glycoprotein (VSG) coat by antigenic variation. VSG switching occurs by frequent homologous recombination, which is thought to require locus-specific initiation. […] Specific association between VSG transcription and replication timing reveals a model for antigenic variation based on replication-derived DNA fragility”.

Second major concern:

It is confounding that a paper built on high-quality data and presenting interesting findings is constructed in a manner that distracts from the findings.

[…]

While the data and discoveries are (and previously were) publication worthy, in my view, the manuscript needs a major rewrite with a change of tone and emphasis on the data that is presented and the specific question the authors aimed to address.

We have rewritten the opening paragraph of the Discussion, more clearly stating our interpretation of the data in the paper and emphasising that we believe endonuclease-mediated DSB formation is unlikely to be the route for VSG switch initiation. We also emphasise the continued uncertainty of exactly what lesion might first form in the active BES to drive antigenic variation:

“Understanding the initiation event(s) of VSG switching by recombination is important, since this element of the reaction may be lineage-specific, and might explain both the elevated rate of the reaction relative to general HR (which catalyses the change in VSG) and the potential focus on the active BES. […] To test the role of endonuclease-generated, BES-focused DSBs in the initiation of VSG switching, we first analysed the genome repair functions of one of two *T. brucei* RecQ-like proteins, TbRECQ2.”

Reviewer #2:

Comments:

*This revised manuscript has improved greatly. The authors have added several key new results and improved the discussions.*

Specifically regarding the authors' conclusion that "phenotypes from RECQ2 null cells argue against models for VSG switch initiation based upon direct DNA break formation", based on the authors' rebuttal and more explanation, this reviewer feels what the authors want to state is that DNA breaks can form due to replication stress, which in turn can induce VSG switching. Therefore, direct DNA break formation, such as a consequence of a specific endonuclease activity, is unlikely the first step of VSG switching. In this regard, this reviewer agrees with the authors. However, this reviewer feels that the term "direct DNA break formation" may not have explained this clearly enough. At this stage, just a few more sentences to clearly describe what do the authors mean about "direct DNA break formation" should suffice.

Please see corrections listed above (to Abstract, Introduction and Discussion), which we hope have clarified the text.

This reviewer appreciates the authors now have performed additional experiments to confirm that when VSG221 is active, it is replicated early, but when it's silenced, it is replicated late. This is a very strong piece of evidence supporting the hypothesis that replication timing is linked with transcription state. The MFA data on VSG121, however, is not as strong. The fact that there are several genomic copies of VSG121 and the fact that VSG121 may not be the sole active VSG among the analyzed population make it hard to interpret the data. It is probably better not to extrapolate too much.

We have stated the following, which we hope sufficiently indicates the uncertainty of BES transcription, or at least VSG121 replication timing, in the BES1 TetR blockade cells:

“Despite the potentially confounding effect of VSG121 array copies that may be late replicating, as well as uncertainty about the transcriptional status of all the VSG BES in these cells, these data suggest earlier replication of VSG121 when the VSG221-containing BES1 is silenced and BES3 is at least one of the BES expressed in the *T. brucei* population”.